# The RNA binding protein FgRbp1 regulates specific pre-mRNA splicing via interacting with U2AF23 in *Fusarium*

Minhui Wang[1], Tianling Ma[1], Haixia Wang[1], Jianzhao Liu [2], Yun Chen [1✉], Won Bo Shim[3✉] & Zhonghua Ma [1✉]

Precursor messenger RNA (pre-mRNA) splicing is an essential and tightly regulated process in eukaryotic cells; however, the regulatory mechanisms for the splicing are not well understood. Here, we characterize a RNA binding protein named FgRbp1 in *Fusarium graminearum*, a fungal pathogen of cereal crops worldwide. Deletion of FgRbp1 leads to reduced splicing efficiency in 47% of the *F. graminearum* intron-containing gene transcripts that are involved in various cellular processes including vegetative growth, development, and virulence. The human ortholog RBM42 is able to fully rescue the growth defects of ΔFgRbp1. FgRbp1 binds to the motif CAAGR in its target mRNAs, and interacts with the splicing factor FgU2AF23, a highly conserved protein involved in 3′ splice site recognition, leading to enhanced recruitment of FgU2AF23 to the target mRNAs. This study demonstrates that FgRbp1 is a splicing regulator and regulates the pre-mRNA splicing in a sequence-dependent manner in *F. graminearum*.

---

[1] State Key Laboratory of Rice Biology, the Key Laboratory of Molecular Biology of Crop Pathogens and Insects, Institute of Biotechnology, Zhejiang University, Hangzhou, China. [2] MOE Key Laboratory of Macromolecular Synthesis and Functionalization, Department of Polymer Science and Engineering, Zhejiang University, Hangzhou, China. [3] Department of Plant Pathology and Microbiology, Texas A&M University, College Station, TX, USA. ✉email: chenyun0927@zju.edu.cn; wbshim@tamu.edu; zhma@zju.edu.cn

In eukaryotic RNA processing, pre-mRNA splicing is a crucial step where introns are removed and exons are joined to form mature mRNAs[1]. In addition, pre-mRNAs are also subject to alternative splicing to generate more than one mature mRNAs from a single gene. Both constitutive and alternative pre-mRNA splicing are catalyzed by a large ribonucleoprotein complex called the spliceosome, which is composed of five core small nuclear ribonucleoprotein (snRNP) particles named U1, U2, U4, U5, and U6, along with many associated protein cofactors[2]. Assembly of the spliceosome occurs in a stepwise fashion. First, U1 snRNP, splicing factor 1 (SF1), and U2 auxiliary factor (U2AF) recognize the 5' splice site (5' SS), branch point sequence (BPS) and 3' splice site (3' SS), respectively, and assemble into an early spliceosome (E complex). Subsequently, SF1 is displaced by the U2 snRNP to form the pre-spliceosome (A complex), which associates with the preformed U4/U6.U5 tri-snRNP to generate the pre-B complex where the tri-snRNP is not yet stably bound. Next, the disruption of the U1/5' SS helix allows stable tri-snRNP integration and U1 snRNP destabilization, yielding the pre-catalytic spliceosome (B complex). The B-complex is then converted into an activated spliceosome (B$^{act}$ spliceosome), which no longer contains U1 and U4. Subsequent remodeling transforms B$^{act}$ into a catalytically competent B* complex (i.e., catalytically activated spliceosome), which catalyzes the first step of splicing, yielding the C complex. The resulting C complex is then remodeled into a step II catalytically activated spliceosome (C* complex), which performs the second catalytic step to produce mature mRNAs[3,4].

The U2AF heterodimer consists of two subunits that are conserved from fission yeast to humans (U2AF65 and U2AF35 in humans and U2AF59 and U2AF23 in fission yeast). The large subunit U2AF65 binds the polypyrimidine tract (Py-tract) immediately downstream of the BPS, while U2AF35 directly comes in contact with AG dinucleotide of the 3' SS to further guide U2AF65 to those relatively weaker Py-tracts[5]. Mutations in U2AF lead to serious diseases, including the Myelodysplasia Syndrome (MDS)[6–8]. Recently, high-throughput RNA sequencing (RNA-seq) assays showed that mutations in U2AF35 alter haematopoiesis due to the mis-splicing of hundreds of gene transcripts resulting from changes in 3' SS recognition[9,10]. In addition to U2AF, mutations in other spliceosome components and splicing regulatory factors are also responsible for various human illnesses, including genetic diseases, neurodegenerative disorders, and cancer[11].

*Fusarium graminearum* is the predominant causal agent of Fusarium head blight (FHB), an economically devastating disease of small grain cereal crops worldwide[12]. In addition to yield losses caused by FHB, mycotoxins such as deoxynivalenol (DON) and zearalenone produced by the pathogen in infested grains seriously threaten the health of humans and animals[13,14]. It would not be difficult to anticipate that pre-mRNA splicing is critical for various cellular processes in *F. graminearum* since the genome of this pathogen contains 77% intron-bearing genes, and 98% of them are constitutively spliced[15,16]. A recent study in *F. graminearum* revealed that FgPrp4, the only kinase among the spliceosome components, regulates the intron splicing in over 60% of gene transcripts by phosphorylation of other spliceosome components. The FgPrp4 mutant showed severe growth defects and a total loss of pathogenicity on wheat[17]. Moreover, the serine/arginine-rich splicing factor FgSrp1, which interacts with FgPrp4 in vivo, also regulates the intron splicing of a subset of genes involved in fungal development and infection[18]. Consistent with these findings, the splicing factor Num1 in *Ustilago maydis* is required for the polarized growth of fungal hyphae, and affects the cell cycle and cell division[19]. In spite of recent reports highlighting the critical roles of pre-mRNA splicing, the mechanisms by which the splice sites are identified and regulated are still largely unknown in fungi.

RNA-binding proteins (RBPs) are critical for co-transcriptional and post-transcriptional gene expression in eukaryotic organisms. Many RBPs are characterized by the presence of various RNA-binding domains including the RNA recognition motif (RRM), K-homology (KH) domain, the zinc finger domain, the DEAD-box domain, and the Pumilio/FBF domain[20]. The *F. graminearum* genome encodes more than 100 predicted RBPs with these domains, but the detailed functions of these RBPs remain uninvestigated. In this study, we identify an uncharacterized RBP (named FgRbp1 hereafter) containing a single RRM domain in *F. graminearum*. The gene deletion mutant displays severe defects in vegetative growth, sexual reproduction and virulence. Using a combination of genetics, biochemistry and high-throughput sequencing approaches, we learned that FgRbp1 is involved in the regulation of pre-mRNA splicing by interacting with a small subunit of U2AF. Moreover, RNA affinity selection assay indicated that FgRbp1 can enhance the recruitment of FgU2AF23 to its target mRNAs. RNA immunoprecipitation followed by deep sequencing (RIP-seq) assay identified 219 putative pre-mRNA targets for FgRbp1 that are involved in various cellular processes, and FgRbp1 directly binds the CAAGR motif deposited in the pre-mRNA targets. Phylogeny analysis showed that orthologs of FgRbp1 are highly conserved in diverse eukaryote lineages. Consistently, we found that the human ortholog RBM42 can rescue the growth defects in ΔFgRbp1. Results from our study provide a strong argument that FgRbp1 orthologs can regulate pre-mRNA splicing in *F. graminearum*, and likely in diverse eukaryotic organisms.

## Results

**Identification of the RNA-binding protein FgRbp1 in *F. graminearum*.** The RRM is one of the most abundant RNA-binding domains in RBPs. The *F. graminearum* genome contains 67 RBPs with the RRM domain (Supplementary Data 1). As listed in Supplementary Data 1, there are ten proteins with no functional implications while the remaining RBPs are annotated. Among these ten RBPs, nine of these proteins have no other characteristic domains except the RRM domain, and Fg_13028 contains a RRM and a Pumilio domain (Supplementary Fig. 1a). Therefore, we were interested in characterizing these uncharacterized RBPs. To initiate our study, we carried out gene deletion for each of these uncharacterized RBPs using homologous recombination strategy. At least three deletion mutants for each gene were obtained by transforming the wild-type strain PH-1, and colony morphologies of these mutants were shown in Supplementary Fig. 1b. Among the ten RBP deletion mutants, ΔFg11902 strain exhibited a severe growth defect, and we designate the putative encoded protein as FgRbp1. The ΔFg11902 strain (herein ΔFgRbp1) was further verified by Southern blot assays (Supplementary Fig. 2). The growth defects of ΔFgRbp1 were restored by genetic complementation with the full-length *FgRBP1* gene in the complemented strain ΔFgRbp1::FgRbp1-GFP (Fig. 1a).

The *FgRBP1* gene is predicted to encode a 351-amino-acid protein containing one RRM domain, which is franked by extended N- and C- terminal tails (Supplementary Fig. 1a). Phylogenetic analysis indicates that FgRbp1 orthologs are conserved in fungi, metazoa and plants, but not in *Saccharomyces* (Supplementary Fig. 3). Amino acid alignment showed that the RRM domain shares a high level of similarity, while the N- and C-terminus vary significantly among different species (Supplementary Fig. 4). Since the RRM of FgRbp1 is similar to that of human counterpart (RBM42) with 51% amino acid identity, we generated RBM42-GFP construct with fungal constitutive promoter RP27 and transformed into ΔFgRbp1 to test whether FgRbp1 and RBM42 are functionally conserved. Significantly, RBM42-GFP fully restored the growth defects in ΔFgRbp1 to the

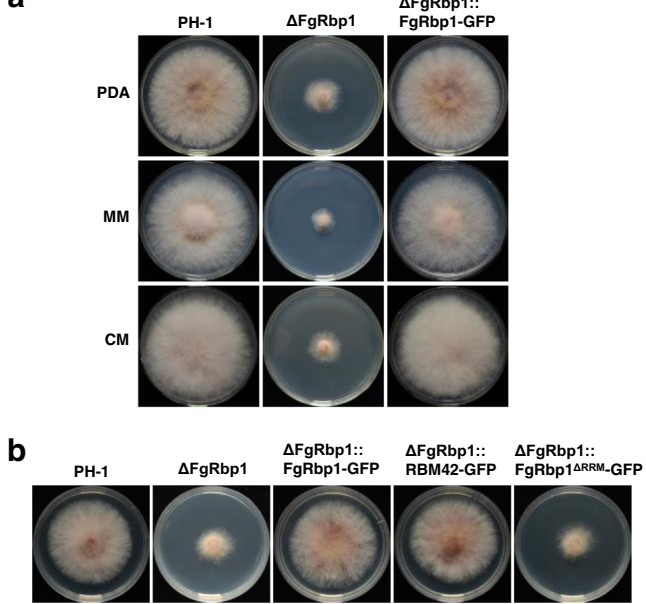

**Fig. 1 Identification of the RRM-containing protein FgRbp1 in *Fusarium graminearum*. a** Colony morphology of PH-1, the FgRbp1-deletion mutant (ΔFgRbp1), and the complemented strain (ΔFgRbp1::FgRbp1-GFP) on potato dextrose agar (PDA), complete medium (CM), and minimal medium (MM) at 25 °C for 3 d. **b** Heterologous complementation with human protein RBM42 and functional analysis of the RRM domain in FgRbp1. The construct expressing human RBM42 and FgRbp1$^{ΔRRM}$ were independently transformed into ΔFgRbp1. The resulting strains ΔFgRbp1::RBM42-GFP and ΔFgRbp1::FgRbp1$^{ΔRRM}$-GFP, along with the control strains ΔFgRbp1::FgRbp1-GFP, ΔFgRbp1 and PH-1 were cultured on PDA at 25 °C for 2 d.

wild-type level (Fig. 1b), indicating FgRbp1 orthologs are functional conserved across eukaryotic kingdoms.

To determine the significance of RRM domain in FgRbp1, the FgRbp1$^{ΔRRM}$-GFP fusion construct (a truncated FgRbp1 lacking RRM domain fused with GFP) was generated and transformed into ΔFgRbp1. In addition, the wild-type FgRbp1-GFP was used as a positive control. As shown in Fig. 1b, the resulting ΔFgRbp1::FgRbp1$^{ΔRRM}$-GFP showed similar growth defects as ΔFgRbp1, indicating that the RRM domain is essential for the biological function of FgRbp1.

**FgRbp1 localizes to the nucleus and directly interacts with U2 snRNP auxiliary factor FgU2AF23**. To gain insight into the biological function of FgRbp1, we first analyzed the subcellular localization of FgRbp1 in *F. graminearum*. The full-length FgRbp1 fused with GFP under the constitutive promoter RP27 was transformed into ΔFgRbp1 to generate a complemented strain ΔFgRbp1::FgRbp1-GFP (the strain as shown in Fig. 1a). Using this complemented strain, we found that FgRbp1-GFP localizes in the nucleus (Fig. 2a, left panel). As anticipated, the RBM42-GFP expressed in ΔFgRbp1 also localizes in the nucleus (Supplementary Fig. 5). In addition, although the RRM domain is essential for the biological function of FgRbp1, deletion of RRM does not alter the nuclear localization of FgRbp1 (Fig. 2a, right panel).

To further elaborate the molecular function of FgRbp1, we screened a *F. graminearum* cDNA library with the yeast two-hybrid (Y2H) approach using FgRbp1 as the bait and identified 87 potential FgRbp1-interacting proteins (Supplementary Data 1). One of these proteins was FgU2AF23 (the ortholog of human U2AF35), which is essential for the determination of 3' splice sites

in pre-mRNA splicing[5]. Given that FgRbp1 is an RNA-binding protein that may participate in pre-mRNA processing, we were interested in this potential interaction. To confirm this interaction between FgRbp1 and FgU2AF23, we first cloned the full-length cDNA sequence of FgU2AF23 into prey plasmid pGADT7 and subsequently validated this interaction using the Y2H assay (Fig. 2b). Concurrently, Y2H assays with a series of truncated FgRbp1 constructs revealed that the uncharacterized N-terminal domain is essential for FgRbp1-FgU2AF23 interaction, while the deletion of the RRM domain weakened the interaction (Fig. 2b). Since the N-terminal domain of FgRbp1 is necessary for the interaction, we therefore generated a FgRbp1$^{ΔN}$-GFP complemented construct (a truncated FgRbp1 lacking N-terminal domain [1-213 aa] fused with GFP) and transformed into ΔFgRbp1 to investigate the function of the uncharacterized N-terminus. The resulting strain ΔFgRbp1::FgRbp1$^{ΔN}$-GFP displayed reduced mycelial growth on PDA, although the growth defect was not as severe as ΔFgRbp1 (Fig. 2c). FgRbp1 and FgU2AF23 interaction, along with the fact that U2AF23 and U2AF59 form a heterodimer complex, led us to ask whether FgRbp1 interacts with FgU2AF59. The result of Y2H assay showed that FgRbp1 does not interact with FgU2AF59 (Supplementary Fig. 6).

Next, subcellular localization assay showed that FgU2AF23 colocalizes with FgRbp1 in the nucleus, revealing the spatial position of their interaction (Fig. 2d). Moreover, we carried out a co-immunoprecipitation (Co-IP) experiment, and the result further verified the interaction between FgRbp1 and FgU2AF23 (Fig. 2e). To test whether the interaction is mediated by RNA, we treated the protein extracts for Co-IP with RNase A prior to immunoprecipitation. Notably, the FgRbp1-FgU2AF23 interaction was RNase A insensitive (Fig. 2e), indicating that this interaction is RNA-independent. We also conducted bimolecular fluorescence complementation (BiFC) assay to further confirm the interaction. As shown in Fig. 2f, the recombined YFP fluorescence signals from FgRbp1-YFP$^{N-terminal}$ and FgU2AF23-YFP$^{C-terminal}$ showed co-localization with the RFP signals of nuclear marker histone 1 (H1), suggesting that they interacted with each other in the nucleus in vivo. Taken together, these results convincingly demonstrated that FgRbp1 physically interacts with FgU2AF23 in *F. graminearum*.

**FgRbp1 is critical for intron splicing efficiency**. U2AF23, the small subunit of U2AF heterodimer, is essential for the determination of 3' splice sites in pre-mRNA splicing. The interaction of FgRbp1 with FgU2AF23 suggests that FgRbp1 participates in pre-mRNA splicing. To test this, we performed RNA-seq in the wild-type and ΔFgRbp1 mutant strains with three independent biological replicates. To evaluate the splicing efficiency of introns, we calculated the intron retention rate (the ratio of normalized read number within introns to normalized read number of exon regions for each intron-containing gene) of each measured intron based on a previously established approach[17,19], and found that the intron retention rate in ΔFgRbp1 is significantly higher than that in the wild type ($p < 2.2e-16$) (Fig. 3a). In ΔFgRbp1, 8689 introns (accounting for 70% introns detected) showed the intron retention rate twofold higher than those in the wild type (corresponding to 4849 genes that are defined as intron retention genes hereafter) (Supplementary Data 3a). Significantly, 52% of these introns have over fourfold increase in intron retention rate (Fig. 3b, Supplementary Data 3a). These results indicate that FgRbp1 regulates pre-mRNA splicing in *F. graminearum*.

**Identification of FgRbp1 targets by RIP-seq**. Since FgRbp1 contains a RRM domain, we questioned whether FgRbp1

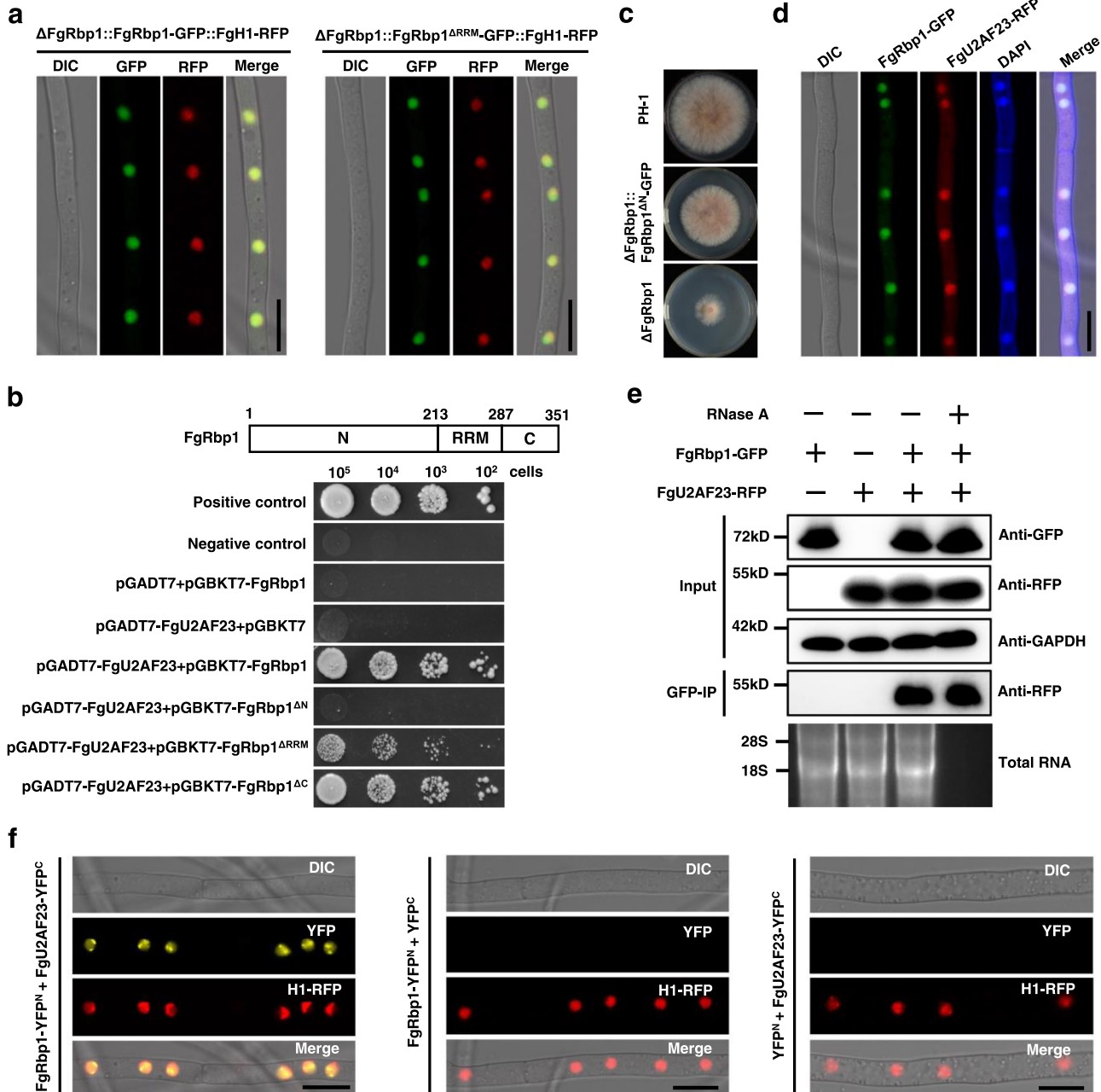

**Fig. 2 Nuclear localization of FgRbp1 and its physical interaction with the splicing factor FgU2AF23. a** Subcellular localization of FgRbp1 (left panel) and FgRbp1$^{\Delta RRM}$ (right panel) in *F. graminearum*. Both FgRbp1-GFP and FgRbp1$^{\Delta RRM}$-GFP localize to the nucleus and co-localize with the RFP-tagged histone H1. Bar = 10 μm. **b** Yeast two hybrid assay verifies interaction of FgRbp1 with FgU2AF23, demonstrating the essential role of the N-terminal domain of FgRbp1 in the interaction. The domain arrangement within FgRbp1 and the number of amino acids of corresponding domains are depicted in the schematic at the top. Tenfold serial dilutions of yeast cells transferred with the bait (a series of truncated FgRbp1) and prey construct (pGADT7-FgU2AF23) were assayed for growth on SD-Leu-Trp-His-Ade plates. A pair of plasmids, pGBKT7-53 and pGADT7-T was used as the positive control, while pGBKT7-Lam and pGADT7-T was used as the negative control. **c** Colony morphology of PH-1, ΔFgRbp1, and ΔFgRbp1::FgRbp1$^{\Delta N}$-GFP on PDA at 25 °C for 3 d. **d** FgRbp1-GFP co-localizes with FgU2AF23-RFP in the nucleus. Nuclei in hyphae were stained with 4′,6-diamidino-2-phenylindole (DAPI). Bar = 10 μm. **e** The interaction between FgRbp1 and FgU2AF23 is confirmed by co-immunoprecipitation assay. The sensitivity of the interaction to RNase A was determined by treating the lysates with RNase A prior and during immunoprecipitation. To assess RNase A treatment efficiency, the RNA from the 1/10 volume lysate of each strain was extracted with Trizol reagent after immunoprecipitation and assessed on the agarose gel. **f** Bimolecular fluorescence complementation (BiFC) analysis verifies the FgRbp1-FgU2AF23 interaction in vivo. Transformants with FgRbp1-YFP$^N$ and FgU2AF23-YFP$^C$ constructs were examined under the confocal microscope. The strains bearing a single construct (FgRbp1-YFP$^N$ or FgU2AF23-YFP$^C$) were used as negative controls. The RFP-tagged histone H1 was used to visualize the nucleus. Bar = 10 μm.

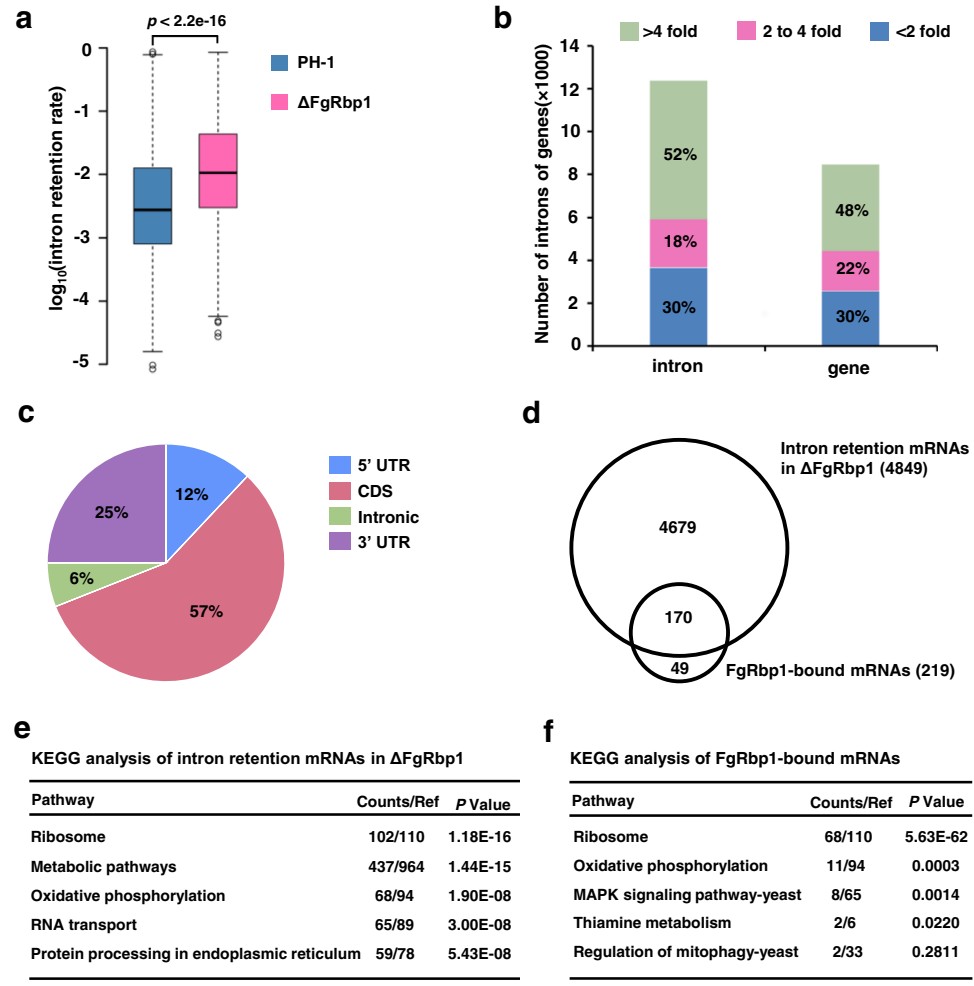

**Fig. 3 Effects of FgRbp1 on pre-mRNA splicing and the identification of target RNAs of FgRbp1 by the RIP-seq assay. a** Box plot of intron retention rates of PH-1 and ΔFgRbp1. Data are represented as boxplots where the middle line is the median, the lower and upper hinges correspond to the first and third quartiles, the upper whisker extends from the hinge to the largest value no further than 1.5 × IQR from the hinge (where IQR is the interquartile range) and the lower whisker extends from the hinge to the smallest value at most 1.5 × IQR of the hinge, while data beyond the end of the whiskers are outlying points that are plotted individually. Statistical significance for the comparison was analyzed by two-tailed *t* test (*P* < 2.2e-16). **b** The percentage of introns and genes with the three marked intron retention rates in ΔFgRbp1 compared with the wild-type PH-1. **c** Distribution of FgRbp1-bound peaks within protein-coding gene bodies divided into 5′-untranslated regions (UTRs), coding sequences (CDSs), 3′-UTRs and intronic regions. **d** Venn diagram showing the overlap between intron retention mRNAs in ΔFgRbp1 and FgRbp1-bound mRNAs. **e** KEGG analysis of intron retention mRNAs in ΔFgRbp1. **f** KEGG analysis of FgRbp1-bound mRNAs identified in RIP-seq.

regulates pre-mRNA splicing via binding directly to its target pre-mRNAs. To test this, we performed the RNA immunoprecipitation (RIP) assay followed by sequencing (RIP-seq) using ΔFgRbp1::FgRbp1-FLAG strain (a complementary strain expressing FLAG-tagged FgRbp1 in ΔFgRbp1). By RIP-seq, we identified 384 FgRbp1 binding peaks (*P* < 0.0005, fold change >1.5), of which 272 peaks were corresponding to 219 protein-coding genes (Supplementary Data 4a), and the remaining peaks were mapped to noncoding RNAs. We analyzed the distribution of these peaks along protein-coding transcripts, and found that more than half of these peaks reside in the coding sequences and 25% and 12% of the peaks reside in the 3′-untranslated regions (UTRs) and 5′-UTRs, respectively, and the remaining peaks (6%) reside in intronic regions (Fig. 3c).

To explore the relationship between RNA binding and pre-mRNA splicing associated with FgRbp1, we overlapped FgRbp1-bound mRNAs with the intron-retention mRNAs in ΔFgRbp1 and found that 170 out of 219 FgRbp1-bound mRNAs (78%) have the intron splicing defect in their pre-mRNAs (Fig. 3d). This suggests a tight relationship between FgRbp1 binding and intron splicing. In addition, pathway analysis using KEGG database revealed similar enrichment pathways in these two classes of genes. They both showed significant enrichment in ribosome and oxidative phosphorylation pathways, which are the top three pathways in enrichment analysis (Fig. 3e, f), further implying that the binding of FgRbp1 to mRNAs and pre-mRNA splicing are closely associated.

**FgRbp1 binds the CAAGR *cis* element in target mRNAs.** To identify the FgRbp1-binding *cis*-element in the target mRNAs, we performed an unbiased search for consensus motifs enriched in peak sequences from 219 potential target pre-mRNAs (corresponding to 272 peaks) by using the DREME algorithm in the MEME suite[21]. The results showed that the most significantly enriched motif is CAAGR (*e*-value = 9.2e–010) (Fig. 4a), which is present in ~70% of the peaks corresponding to 75% of transcripts (Supplementary Data 4a). To verify the interaction of FgRbp1 with this *cis* element, we carried out electrophoretic mobility shift assays (EMSAs) by using purified GST-FgRbp1 fusion protein (Fig. 4b) and the synthetic RNA of six repeats of 5′-CAAGA-3′ or

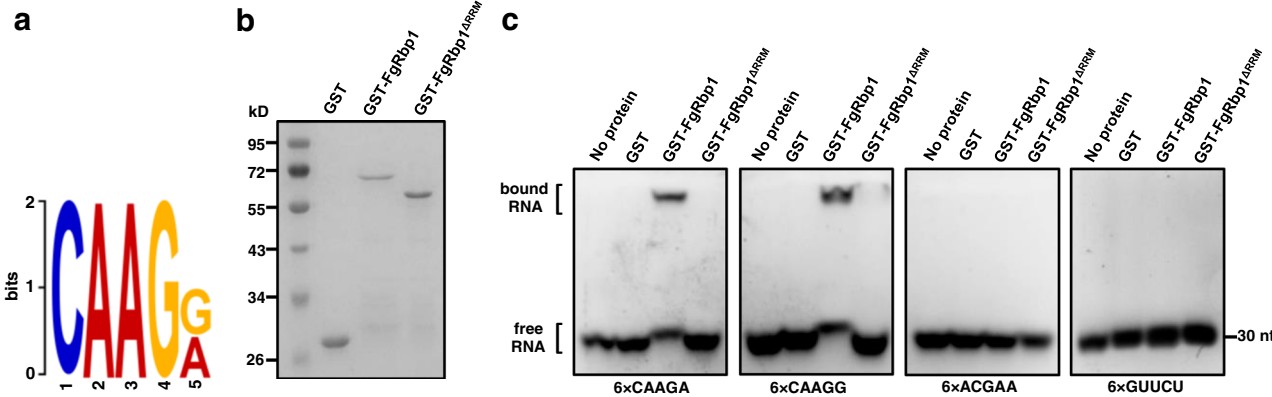

**Fig. 4 FgRbp1 binds the CAAGR cis element. a** Sequence logo of the enriched motif among the peak sequences of FgRbp1 target genes. The y axis (letter size) indicates the information content (bits) of each position. **b** GST-FgRbp1 and GST-FgRbp1$^{\Delta RRM}$ were expressed in *E. coli*, purified using glutathione sepharose, resolved by SDS-PAGE on a 12.5% acrylamide gel, and stained with coomassie blue. The size of relevant molecular weight markers (MWM) are indicated on the left. **c** In vitro binding of FgRbp1 to the CAAGR motif in the EMSA assay. Synthetic six repeats of 5′-CAAGA-3′, 5′-CAAGG-3′, 5′-ACGAA-3′ and 5′-GUUCU-3′ sequences were incubated alone or with GST, GST-FgRbp1, or GST-FgRbp1$^{\Delta RRM}$ for 5 min at room temperature and separated on a 2.0% agarose gel.

5′-CAAGG-3′. As shown in Fig. 4c, retardation of RNA migration was observed for both 5′-CAAGA-3′ and 5′-CAAGG-3′ repeated motif, whereas GST or GST-FgRbp1$^{\Delta RRM}$ (a truncated FgRbp1 lacking RRM domain) was unable to bind to the motif at all. As a control, FgRbp1 was unable to bind to a scrambled RNA sequence 5′-ACGAA-3′, nor a mutated RNA sequence 5′-GUUCU-3′ with six repeats (Fig. 4c), indicating the binding specificity of FgRbp1 to the CAAGR motif. Together, these results indicate that the CAAGR motif deposited in the target mRNAs is one of *cis*-elements for FgRbp1 and that the RRM domain is essential for the binding of FgRbp1 to this *cis*-element.

**FgRbp1 binds and regulates splicing of ribosomal protein genes.** Among the 170 overlapped mRNAs in FgRbp1-bound mRNAs and intron retention mRNAs, 68 mRNAs encode ribosomal proteins (Fig. 3e, f). Sequence analysis demonstrated that peaks of 64 target ribosomal protein genes (RPGs) contain the CAAGR motif (Supplementary Data 4b), thereby the motif may direct the specificity of FgRbp1 to these RPGs. Ribosomal proteins (RP) are fundamental components of the ribosome, contributing to the assembly of the ribosome and its ability to synthesize proteins. Inactivation of RP has been linked to developmental abnormalities in a variety of human organ systems, including short stature, craniofacial defects, thumb malformation and heart defects[22]. Similarly, these essential RPGs play crucial roles in mycelium development in *F. graminearum* based on the fact that we failed to generate any viable RP deletion mutants. Based on these outcomes, we conclude that the collective splicing defects of these RPGs are contributing to the severe growth defects in ΔFgRbp1.

To verify the splicing defect in these RPGs observed by RNA-Seq, we measured the intron splicing ratio (the ratio of spliced RNA to unspliced RNA) in nine RPGs randomly selected from 68 using the qRT-PCR approach described previously[19,23,24] (for details, see "Methods"). Since each RPG has multiple introns with splicing defects, we then chose the intron with the highest retention rate in each RPG for verification. As shown in Fig. 5a, the splicing ratios of nine RPGs decreased dramatically in ΔFgRbp1 when compared with those of the wild-type strain, while the complemented strain ΔFgRbp1::FgRbp1-GFP bearing full length FgRbp1 restored the splicing ratios to the wild-type levels. Meanwhile, the RRM-depleted complemented strain (ΔFgRbp1::FgRbp1$^{\Delta RRM}$-GFP) showed similar splicing defects

as ΔFgRbp1, indicating that the RRM domain is necessary for the splicing regulation of FgRbp1 (Fig. 5a). The N-terminal domain-depleted complemented strain (FgRbp1::FgRbp1$^{\Delta N}$-GFP) also showed decreased splicing ratio compared with PH-1, although the decline in the complemented strain was not as great as that in ΔFgRbp1 (Fig. 5a), suggesting that the N-terminal domain also plays a role in splicing regulation for FgRbp1. Consistently, the abundances of two ribosomal proteins RPS20 and RPS3 in ΔFgRbp1 were reduced to 25% and 42% of wild-type level, respectively (Fig. 5b). Moreover, qRT-PCR assays also demonstrated that the gene expression levels of all nine tested genes are not significantly altered in ΔFgRbp1 when compared to those in the wild-type strain (Fig. 5c), suggesting the decrease in protein abundance is not due to the decrease in transcription, but rather due to the ineffective pre-mRNA splicing. Moreover, the RNA-seq data also revealed that all 68 RPGs with splicing defects showed no significant fold change in gene expression levels, except for one gene FGSG_06724 that exhibits a 2.4 fold increased expression in ΔFgRbp1 compared with the wild-type strain (Supplementary Data 3b).

Next, the binding of FgRbp1 to the peak sequences of RPGs in vivo were confirmed by RIP-qPCR (Fig. 5d). To validate whether FgRbp1 directly binds to these RPG pre-mRNAs, we carried out EMSAs by using purified GST-FgRbp1 and in vitro transcribed ~250-bp mRNA fragments of three RPGs (*FgRPL25*, *FgRPL39* and *FgRPS20*) containing the peaks identified in the RIP-seq (the in vitro transcribed mRNA sequence contains the sequence confirmed in Fig. 5d). As shown in Fig. 5e, the migration of fragment mRNAs through the gel was obstructed by GST-FgRbp1 whereas no obstruction was observed by GST or GST-FgRbp1$^{\Delta RRM}$. These results indicate that FgRbp1 directly binds to these RPG pre-mRNAs.

**FgRbp1 regulates splicing of the ribosomal protein genes in a sequence-dependent manner.** We next asked if the CAAGR motif in RPGs were required for the effect of FgRbp1 on splicing. Thus, we further used *FgRPS20* as a model to investigate the role of FgRbp1–CAAGR interaction in regulating pre-mRNA splicing. *FgRPS20* contains 5′-CCAAGAACCTCAAGA-3′ sequence (having two repeated 'CAAGA' sequences) located at the downstream exon of the first intron (127 nt downstream of intron1) (Fig. 5f). We then designed two constructs, the wild-type construct containing exon1-intron1-exon2 of *FgRPS20* fused with GFP and the

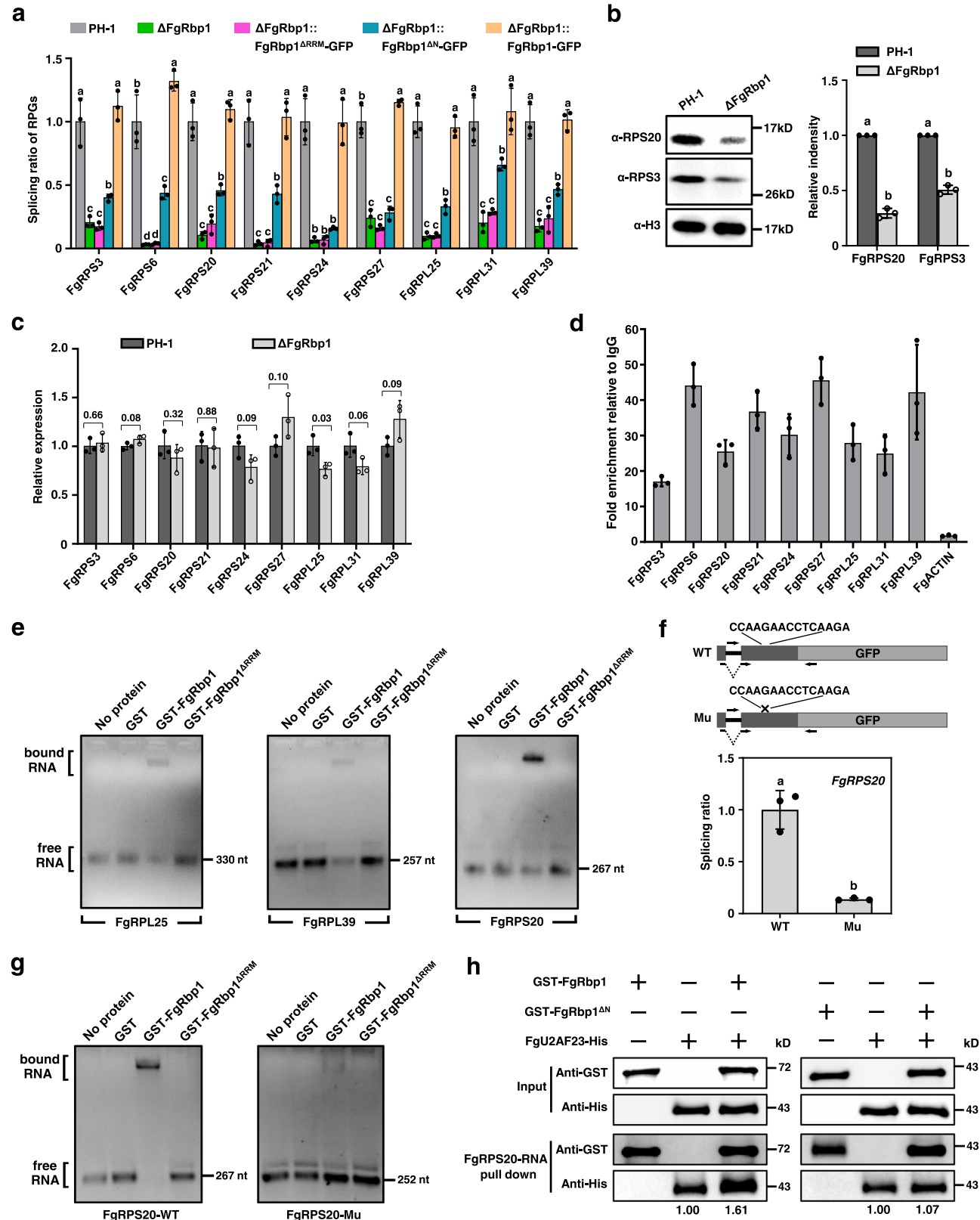

mutated construct which is consistent to the wild-type construct except for the deletion of 5'-CCAAGAACCTCAAGA-3' sequence (Fig. 5f). Since *FgRPS20* is predicted to be essential for fungal growth (i.e., we were unable to obtain *FgRPS20* deletion mutant after extensive effort), we transformed the wild-type strain PH-1 with each construct and assayed the splicing ratio of intron 1 in these transformants by qRT-PCR. As shown in Fig. 5f, the splicing ratio of *FgRPS20* intron 1 was reduced dramatically in the strain containing the mutated construct when compared to that of the wild-type construct. To further confirm that the splicing defect of *FgRPS20* intron 1 in the mutated-construct containing strain resulted from the loss of binding ability of FgRbp1 to the

**Fig. 5 FgRbp1 regulates the splicing of ribosomal protein genes (RPGs) and facilitates the recruitment of FgU2AF23 to target mRNAs. a** qRT-PCR validation of the splicing defects of introns in ribosomal protein genes. Data are presented as means ± standard deviations (SD), $n = 3$ biologically independent replicates. Different letters indicate a significant difference ($P < 0.05$) according to one-way ANOVA followed by Fisher's least significant difference (LSD) test. **b** Comparison of the protein amount of RPS3 and RPS20 in the wild-type PH-1 and ΔFgRbp1 strains. The intensities of the western blot bands were quantified with the program IMAGE J. Error bars represent SD ($n = 3$ independent experiments). **c** Relative gene expression level of nine RPGs in PH-1 and ΔFgRbp1. Error bars represent SD ($n = 3$ biologically independent replicates). Individual $P$ values are denoted above the comparison lines. **d** Verification of FgRbp1 binding on the mRNAs of RPGs by RIP-qPCR. *ACTIN* served as a control. Data are presented as means ± SD ($n = 3$ biologically independent replicates). **e** EMSA confirmation of the direct binding of FgRbp1 to the mRNAs of RPGs. In vitro transcribed RNAs containing the peak sequence were incubated with purified proteins described in Fig. 4b. **f** qRT-PCR analysis of the splicing ratio of *FgRPS20* in the strains transformed with WT or mutated construct *FgRPS20*. PH-1 was transformed with either the wild-type construct (WT) containing exon1-intron1-exon2 of *FgRPS20* fused with GFP or the mutated construct (Mu) with the sequence 5′-CCAAGAACCTCAAGA-3′ downstream of the intron deleted. The schematic of the two constructs of *FgRPS20* is at the top. The arrows indicate the primer locations used to measure the splicing ratio of introns. **g** EMSA assay to test the binding ability of FgRbp1 to mRNA sequence "*FgRPS20*-WT" or "*FgRPS20*-Mu". The RNA sequence of *FgRPS20*-WT is identical to that of exon2 (267-bp) as indicated in **d** and **e**, and *FgRPS20*-Mu is consistent to the *FgRPS20*-WT except that the motif 5′-CCAAGAACCTCAAGA-3′ was deleted. **h** RNA affinity selection assay. The biotin-labeled *FgRPS20* mRNA were immobilized on streptavidin beads and incubated with single protein: GST-FgRbp1, GST- FgRbp1ᐃN, FgU2AF23⁻His, or pairs of proteins: GST-FgRbp1/ FgU2AF23-His, GST- FgRbp1ᐃN/FgU2AF23⁻His. The bound proteins were detected by western blot with the anti-GST or anti-His antibody. The fold-change in the Anti-His of pull-down for mixed proteins relative to that of FgU2AF23 was indicated below the band. In **b**, **c** and **f**, bars represent means ± SD of three independent experiments. Different letters indicate a significant difference ($P < 0.05$) according to the two-tailed unpaired Student's $t$ test.

mutated mRNA, EMSA experiment was carried out by using two in vitro transcribed mRNA fragments *FgRPS20*-WT and *FgRPS20*-Mu. The sequence of the mRNA fragment *FgRPS20*-WT is identical to that of exon 2 (267-bp) as indicated in Fig. 5f, and *FgRPS20*-Mu is consistent to the *FgRPS20*-WT except that the 5′-CCAAGAACCTCAAGA-3′ sequence in exon 2 was deleted. The results of EMSA showed that FgRbp1 bound to the mRNA sequence of *FgRPS20*-WT, but hardly to the sequence of *FgRPS20*-Mu (Fig. 5g). Taken together, these results indicate that FgRbp1 regulates the splicing of *FgRPS20* intron 1 through direct interaction with the *cis*-element CAAGR.

**FgRbp1 enhances the recruitment of FgU2AF23 to the *FgRPS20* mRNA.** Given that the unknown N-terminal domain of FgRbp1 was required for the interaction with FgU2AF23 (Fig. 2b), and the deletion of N-terminal domain also led to the intron splicing defect (Fig. 5a), it is reasonable to propose that that FgRbp1-FgU2AF23 interaction should contribute to the role of FgRbp1 in splicing. The study of splicing regulation mechanism in human shows that many splicing regulators (e.g., SR protein or other RBPs) can promote exon inclusion or intron splicing by recruiting U1 snRNP to the 5′ splice site or U2 auxiliary factor (U2AF) to the 3′ splice site through protein–protein interactions in early steps of spliceosome assembly[25–27]. Since U2AF35 (the ortholog of FgU2AF23 in human) is an essential protein in the first step of assembly of spliceosome, we hypothesized that FgRbp1-FgU2AF23 interaction also promotes the recruitment of FgU2AF23 to FgRbp1-bound mRNAs to regulate intron splicing. To assess this, RNA affinity selection assays were conducted using *FgRPS20* mRNA as a representative sample. The biotin-labeled *FgRPS20* mRNA fragment containing exon1-intron1-exon2 (as shown in Fig. 5f) was in vitro transcribed and immobilized on streptavidin magnetic beads. The baits were incubated with either in vitro purified GST-FgRbp1 or FgU2AF23-His (Supplementary Fig. 7), or both proteins; while the interaction domain deleted protein GST-FgRbp1ᐃN (Supplementary Fig. 7) was used as a control. As observed in Fig. 5h, GST-FgRbp1 alone bound to the *FgRPS20* RNA baits, which is consistant with the EMSA result in Fig. 5e. Meanwhile FgU2AF23-His alone also bound to the RNA baits, which is in line with the previous reports that U2AF35 (human counterpart) or U2AF23 (fission yeast counterpart) binds to AG dinucleotide at the 3′ splice site in intron[28,29]. However, when two proteins were mixed together and incubated with the RNA baits, the

binding of FgU2AF23-his to the RNA was significantly increased by ~60% (Fig. 5h). Moreover, GST-FgRbp1ᐃN could also alone bind to the *FgRPS20* RNA baits, but the addition of GST-FgRbp1ᐃN did not increase the FgU2AF23-his binding to the RNA baits (Fig. 5h), suggesting that this recruitment is dependent on the interaction between the two proteins. These results indicate that FgRbp1 can enhance the recruitment of FgU2AF23 to the target *FgRPS20* mRNA via interacting with FgU2AF23.

**FgRbp1 regulates the splicing of pre-mRNAs involved in sexual reproduction.** Sexual reproduction is a critical part of *F. graminearum* life cycle, and the perithecia produced from sexual reproduction serve as the key primary inoculum[30]. To characterize the function of FgRbp1 in sexual development, we assayed perithecium formation on carrot agar. As shown in Fig. 6a, deletion of FgRbp1 completely abolishes the production of perithecia, while the wild-type strain PH-1 and the complemented strain ΔFgRbp1::FgRbp1-GFP produce abundant perithecia. To explore the cause of sexual infertility in ΔFgRbp1, we screened for mating related genes from the FgRbp1 target mRNAs and found two pivotal genes *FgMCM1* and *FgGPA1* that have been reported to control *F. graminearum* sexual production[31,32]. The RNA-seq data revealed that two introns of *FgMCM1* and three introns of *FgGPA1* are retained to different extents (Supplementary Data 3a). The qRT-PCR assays substantiated the severe splicing defects of the two genes in ΔFgRbp1 (Fig. 6b), which is in agreement with the loss of perithecium production in the mutant. Meanwhile, the transcription level of *FgMCM1* and *FgGPA1* remained at a similar level in PH-1 and ΔFgRbp1 (Supplementary Fig. 8a). The peak fragments of *FgMCM1* and *FgGPA1* transcripts identified in RIP-seq are both located in the 3′-UTR of these two genes (Fig. 6b). *FgGPA1* contains a CAAGA motif in the identified 200-bp peak fragment, while *FgMCM1* contains no CAAGR motif in the identified 173-bp peak fragment (Supplementary Data 4a). Nevertheless, RIP-qPCR and EMSA confirmed the direct binding of FgRbp1 to the two peak fragments of *FgMCM1* and *FgGPA1* (Fig. 6c, d). To investigate whether the binding of FgRbp1 to 3′-UTR of *FgMCM1* and *FgGPA1* is related to the splicing of these two genes, we constructed strains with a mutated 3′-UTR for *FgMCM1* (the 173-bp peak sequence in 3′-UTR was deleted) and a mutated 3′-UTR for *FgGPA1* (the 200-bp peak sequence in 3′-UTR was deleted) using an in situ replacement strategy. The FgRbp1's binding region in 3′-UTR of *FgMCM1* is responsible for the

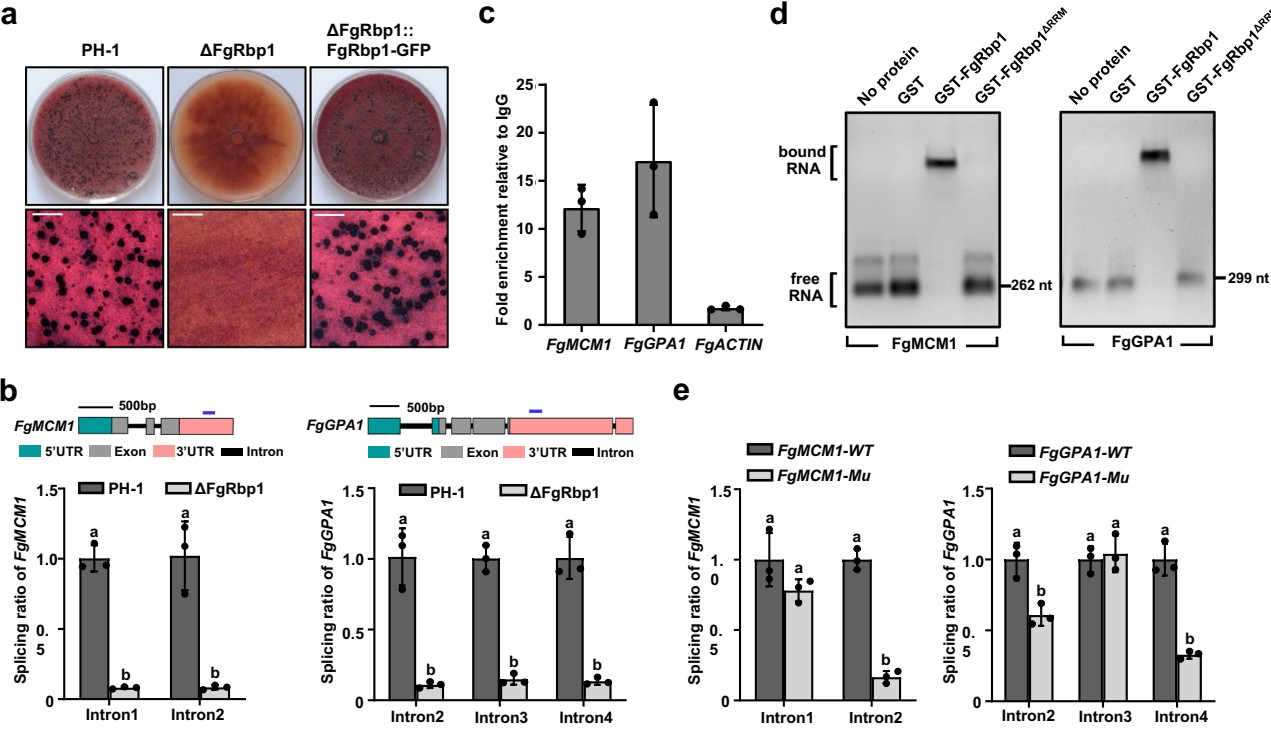

**Fig. 6 FgRbp1 is required for sexual reproduction in *F. graminearum* via regulating the splicing of mating-related genes. a** Deletion of FgRbp1 leads to sexual sterility. Each strain was grown on carrot agar for induction of perithecial formation. Bar = 500 μm. **b** qRT-PCR validation of the splicing defect of introns in *FgMCM1* and *FgGPA1*. The schematic of *FgMCM1* or *FgGPA1* is shown at the top. The blue solid line denotes the binding peak identified in RIP-seq. **c** Verification of FgRbp1 binding on mRNAs of *FgMCM1* and *FgGPA1* by RIP-qPCR. *ACTIN* served as a control. Error bars represent means ± SD of three repeated experiments. **d** EMSA confirmation of the direct binding of FgRbp1 to the RNAs of *FgMCM1* or *FgGPA1*. In vitro transcribed RNAs containing the peak sequence of *FgMCM1* or *FgGPA1* were incubated with GST-fusion proteins described in Fig. 4b. **e** Deletion of the binding fragments in **b** resulted in decreased intron splicing ratio of introns in *FgMCM1* or *FgGPA1*. The wild-type PH-1 strains contain the peak fragment while Mu strains are deleted with the peak fragments. In **b** and **e**, bars represent means ± SD (n = 3 biologically independent replicates). Different letters indicate a significant difference (P < 0.05) according to the two-tailed unpaired Student's *t* test.

splicing of intron 2 of *FgMCM1*, while the 3′-UTR binding region of *FgGPA1* is related to the splicing of intron 2 and intron 4 of *FgGPA1* (Fig. 6e). Together, these results indicate that FgRbp1 is critical for sexual reproduction in *F. graminearum* at least partially through regulating the splicing of two major mating-related mRNAs.

**FgRbp1 is required for full virulence.** The virulence of ΔFgRbp1 was evaluated by point inoculation of conidial suspension on flowering wheat heads. After 15 dpi, ΔFgRbp1 and ΔFgRbp1::FgRbp1^ΔRRM^-GFP failed to infect the spikelets, even in the spikelet directly inoculated, while the wild-type and the complemented strains resulted in extensive scab symptoms on wheat heads (Fig. 7a, left). The strain ΔFgRbp1::FgRbp1^ΔN^-GFP also exhibited dramatically decreased pathogenicity on wheat heads, although the degree of decline was not as severe as ΔFgRbp1 (Fig. 7a, left). Consistently, ΔFgRbp1, ΔFgRbp1::FgRbp1^ΔRRM^-GFP and ΔFgRbp1::FgRbp1^ΔN^-GFP also showed attenuated virulence on corn silks (Fig. 7a, right). In order to verify the pathogenicity defect of ΔFgRbp1 in plants, the *ACTIN* gene fused with RFP was transformed into PH-1 and ΔFgRbp1 to track the growth of fungal hyphae in inoculated and nearby spikelets. After 7 dpi, RFP fluorescence signals were not detected in ΔFgRbp1 while the signals were observed in the wild-type inoculated and nearby spikelets (Fig. 7b). Next, to explore the underlying regulatory mechanisms of FgRbp1 pathogenicity, we performed cellophane penetration assay. As shown in Fig. 7c, ΔFgRbp1 was

unable to penetrate a cellophane sheet. Consistently, ΔFgRbp1 failed to form any penetration structures on inoculated spikelets, whereas the wild-type strain formed a number of penetration structures when observed by electronic scanning microscope (Fig. 7d). Therefore, it is reasonable to conclude that the loss of pathogenicity is due to the defect in penetration capability in ΔFgRbp1.

In plant pathogenic fungi, the Fus3/Kss1 MAPK pathway is well known to control a variety of virulence-related functions, including formation of penetration structure and secretion of plant cell wall degrading enzymes[33,34]. Thus, we asked whether FgRbp1 regulates the splicing of pre-mRNAs that are responsible for host penetration. The transcription factor FgSte12, which is downstream of the Fus3/Kss1 MAPK pathway, was identified as a FgRbp1 target with splicing defect in our RIP-seq and RNA-seq data (Supplementary Data 3, 4). A previous study showed that the *FgSTE12* deletion mutant is unable to form infection structures on wheat[35]. The splicing ratios of the two introns of *FgSTE12* decreased dramatically in ΔFgRbp1 when compared with those in the wild type (Fig. 7e), although the transcription level of *FgSTE12* remained at a similar level in both strains (Supplementary Fig. 8b). RIP-qPCR and EMSA assays confirmed the direct interaction between FgRbp1 and the identified peak fragment in *FgSTE12* exon 1 with a CAAGA motif (Fig. 7f, g). Furthermore, the deletion of the CAAGA motif in the peak fragment led to significantly reduced intron splicing ratio of two introns in *FgSTE12* (Fig. 7h). Taken together, these results indicate that FgRbp1 controls virulence of *F. graminearum* partially by

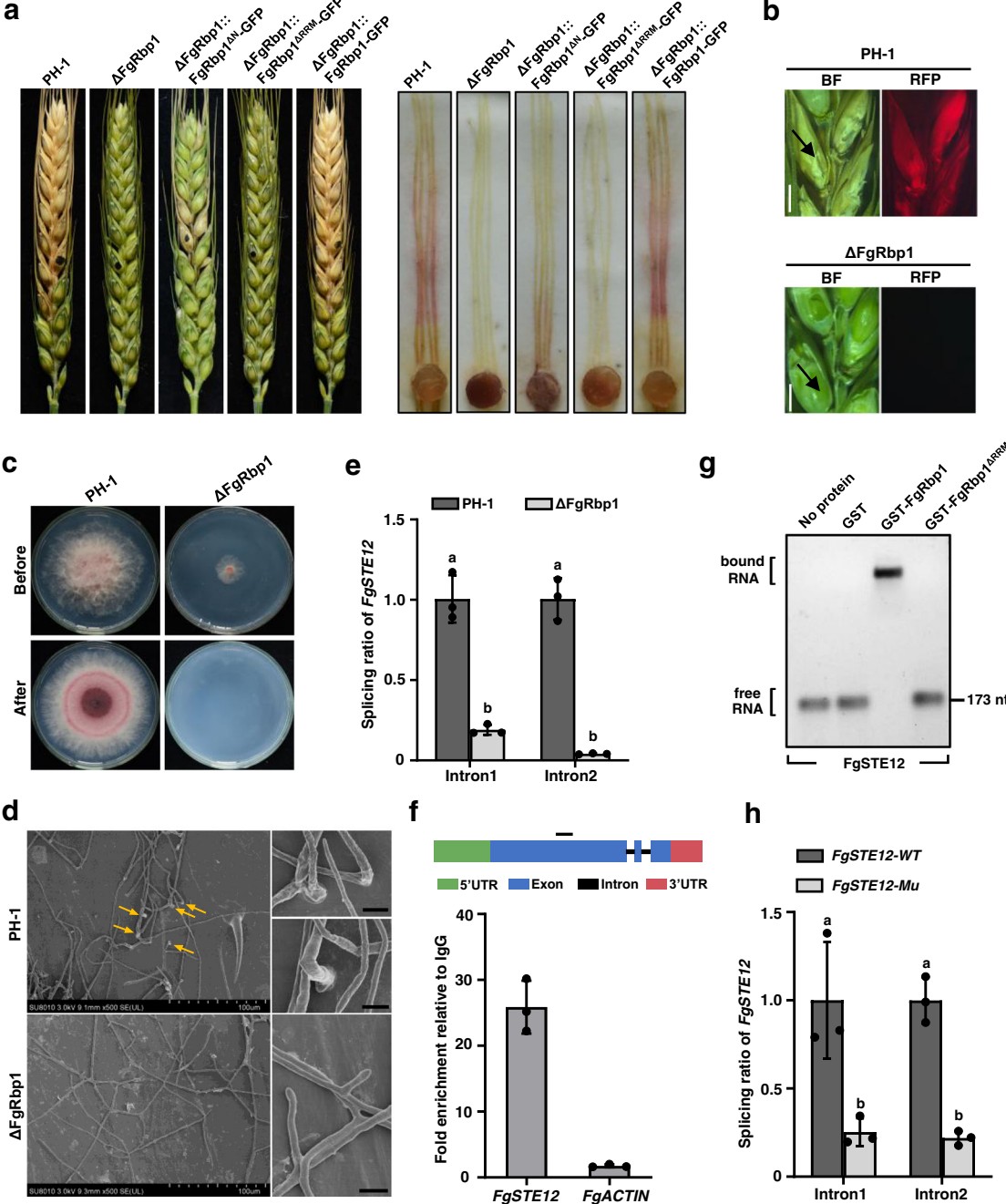

**Fig. 7 FgRbp1 is essential for the formation of fungal penetration structures on wheat. a** Deletion of FgRbp1 leads to significantly reduced virulence on wheat heads (left panel) and corn silks (right panel). Flowering wheat heads were point-inoculated with the PH-1, ΔFgRbp1, ΔFgRbp1::FgRbp1$^{ΔN}$-GFP, ΔFgRbp1::FgRbp1$^{ΔRRM}$-GFP and ΔFgRbp1::FgRbp1-GFP. The infected wheat heads were photographed 14 days post-inoculation (dpi). Infected corn silks were examined 5 dpi. **b** Cross-sections of inoculated and adjacent wheat spikelets. Spikelets were infected with PH-1 or ΔFgRbp1, each strain bearing FgActin-RFP. The samples were taken at 7 dpi. Inoculated spikelets were indicated with black arrows. BF, bright field; RFP, red fluorescent protein. Bar = 200 μm. **c** Penetration of the wild type PH-1 and ΔFgRbp1 into cellophane membrane. After removing colonies of PH-1 and ΔFgRbp1 grown over cellophane membranes on MM (Before) for 3d, the plates were incubated for three additional days to examine for hyphal growth (After). **d** Electronic scanning microscope examination of the penetration structures on wheat glumes. Wheat head spikelets were inoculated with conidia of the wild-type PH-1 or ΔFgRbp1, and sampled after 7 dpi. Penetration structures are marked with yellow arrows. Bar = 5 μm. **e** Introns of *FgSTE12* showed significantly reduced splicing ratio (left panel) in ΔFgRbp1 compared with those in the wild-type PH-1. **f** Verification of FgRbp1 binding on the mRNA of *FgSTE12* by RIP-qPCR. *ACTIN* served as a control. Error bars represent means ± SD of three repeated experiments. The schematic of *FgSTE12* is shown at the top. The black solid line denotes the binding peak identified in RIP-seq. **g** EMSA confirmation of the direct binding of FgRbp1 to the in vitro transcribed mRNAs containing the peak sequence of *FgSTE12*. The in vitro transcribed mRNA sequence is identical to **f**. **h** The deletion of the CAAGA motif in the peak fragment resulted in the decreased intron splicing ratio of two introns in *FgSTE12*. The strain *FgSTE12-WT* contains the CAAGA motif in the peak fragment, while the strain *FgSTE12-Mu* was deleted with the CAAGA motif. In **e** and **h**, bars represent means ± SD (*n* = 3 biologically independent replicates). Different letters indicate a significant difference (*P* < 0.05) according to the two-tailed unpaired Student's *t* test.

regulating pre-mRNA splicing of the transcription factor *FgSTE12*.

## Discussion

Splicing of mRNA is an essential process in eukaryotic gene expression. Recognition of the splicing signals and removal of introns in pre-mRNA are largely carried out by the spliceosome. Although components of spliceosomal complexes (e.g., E, A, B, and C) have been well established in the model organism *S. cerevisiae*, new associated factors are reported occasionally in various eukaryotes[36–38]. In *F. graminearum*, the highly conserved spliceosome component FgPrp4 regulates pre-mRNA splicing likely by phosphorylation of other components of the U4/U6-U5 tri-snRNP[17]. However, Prp4 ortholog is absent in all sequenced *Saccharomycotina* species, except *Yarrowia lipolytica*[39]. Because *Y. lipolytica* has over 1500 introns in its genome, and *S. cerevisiae* only 376 introns, it is reasonable to speculate that *Saccharomycotina* species may have lost the Prp4 ortholog after going through massive intron loss during evolution[17]. In this study, we characterize the RRM-containing protein FgRbp1 in *F. graminearum* and showed that FgRbp1 interacts with an essential splicing factor FgU2AF23 in the nucleus. Deletion of FgRbp1 does not block intron splicing but affects intron splicing efficiency in 47% (4849 out of 10268) of the intron-containing genes in *F. graminearum*. Significantly, the human RBM42 protein is able to completely restore the growth defects in ΔFgRbp1. Similar to Prp4, FgRbp1 ortholog is found in diverse eukaryotes, but not in *S. cerevisiae*. These results indicate that there is a greater level of complexity associated with the regulatory mechanism of pre-mRNA splicing in diverse eukaryotes than currently outlined in the budding yeast.

In human, the FgRbp1 ortholog RBM42 was first identified as a heterogeneous nuclear ribonucleoprotein K (hnRNP K) -binding protein playing a role in the maintenance of cellular ATP level under stress by protecting their target mRNAs[40]. Proteomic assays later revealed that RBM42 may be an integral component of the B complex[41], U4/U6.U5 tri-snRNP[42] and pre-B complex[43], but the role it plays in these complexes is still unknown. More recently, RBM42 has been verified as a component of human tri-snRNP and pre-B spliceosome by cryo-electron microscopy (cryo-EM)[44]. In human tri-snRNP, RBM42 contributes to stabilizing the quasi-pseudoknot (a structure formed by the U4 snRNA nucleotides 63 to 67) by binding to the U4 nucleotides 68 to 70th via the RRM domain. The quasi-pseudoknot brings U6 snRNA nucleotides A36 and A47 closer to enable the U6 ACAGAGA-box sequence to loop out, and project toward the U1 snRNP in the pre-B complex for 5'SS pairing. In addition, a study on the ortholog of RBM42 in apicomplexan protozoans also revealed the conserved function of RBM42 as a component of tri-snRNP complex[38]. In our study, we showed FgRbp1 physically interacts with FgU2AF23 both in vitro and in vivo, a well-characterized splicing factor of early spliceosome complex (E complex). A previous study on protein-protein interactions among human spliceosomal proteins demonstrated that RBM42 interacts with U1-A (core protein of U1 snRNPs) in Y2H and Co-IP assay[45], suggesting that RBM42 is also involved in the early spliceosome assembly in mammalian cells. These results indicate that FgRbp1 orthologs may regulate pre-mRNA splicing at different steps via various interactors.

In this study, we found that 78% of RIP-target genes overlapped with intron retention genes, and that the KEGG analysis revealed similar enriched pathways of these two gene classes (Fig. 3d–f). These results strongly imply that FgRbp1 binds to its target pre-mRNAs to regulate their splicing. However, there are 4849 genes with intron splicing defects, with only 170 genes as RIP target genes (Fig. 3d). Thus, we propose that FgRbp1 has both direct and indirect roles in splicing based on the following arguments. Since FgRbp1 is the functional ortholog of human RBM42, it is reasonable to predict FgRbp1 has the same role as RBM42 in basic spliceosome assembly (e.g., the pre-B complex assembly) via stabilizing the quasi-pseudoknot, thereby having a direct role in regulating global pre-mRNA splicing of many non-target genes. Thus, the splicing of these non-target genes are indirectly regulated by FgRbp1. More importantly, FgRbp1 directly binds and regulates the splicing of 68 transcripts coding for ribosomal proteins as demonstrated in our results (Fig. 5; Supplementary Data 3, 4). The mis-splicing of these ribosomal proteins is likely to lead to a decline in the overall translation ability in cells, subsequently affecting splicing of a wide range of genes.

In current study, we identified a cis element CAAGR in RIP-seq data and confirmed that FgRbp1 directly binds to this motif via EMSA. However, not all peak sequences of target genes contain the CAAGR motif. Among the 219 FgRbp1 target mRNAs identified by RIP assay, 165 mRNAs contained this motif, while the remaining 54 mRNAs (including *FgMCM1*) did not (Supplementary Data 4a). Therefore, we can presume that FgRbp1 resort to other affinity mechanisms for binding to these CAAGR-lacking genes, perhaps via other cis elements or specific RNA secondary structures. We found that most CAAGR binding sites are located within the exons around the intron, with a proportion of sequences located in 3′-UTR (like *FgMCM1* and *FgGPA1*). Theoretically, splicing regulatory elements may function at any location in a pre-mRNA, but most studies focused on 200–300 nucleotides adjacent to the observed splice sites with the most identifiable sequence features[25]. In this study, we found that the CAAGR motif (CCAAGAACCTCAAGA) located in exon2 (127 nt downstream of the adjacent intron1) that bound to FgRbp1 is responsible for the splicing of intron1 (Fig. 5f, g). This is consistant with the intron retention (IR) regulation observed, in human, i.e., specific RNA binding proteins may modulate the level of particular IR events by binding specific motifs in introns or their flanking exons[46]. Several studies showed that remote regulatory sequences (e.g., 3′-UTR) are also important to the splicing outcome via forming RNA secondary structures[47,48]. Further investigation is needed to gain clearer understanding of how FgRbp1 regulates splicing via remote regulatory sequences.

U2AF23, the small subunit of U2AF heterodimer, is conserved among diverse eukaryotes, except in *S. cerevisiae*. Interestingly, *S. cerevisiae* does not contain an ortholog of FgRbp1. In this study, we identified FgRbp1 as an additional interacting partner of the U2AF small subunit (Fig. 2). The amino acid sequence conservation between FgU2AF23 and FgRbp1 suggests that the interaction of the two proteins may be retained in other organisms. Indeed, the RBM42 protein (human counterpart of FgRbp1) interacts with FgU2AF23 in Y2H assay (Supplementary Fig. 9). Here, we provided evidence in vitro that FgRbp1 enhances the recruitment of FgU2AF23 to its target mRNA *FgRPS20*, and this recruitment is dependent on the interaction between the two proteins (Fig. 5h). Furthermore, by comparing the degree of splicing defects between the intron-retention genes excluded from RIP target genes (4679 genes) (refer to as "non-overlapped genes") and those overlapped with RIP target genes (170 genes) (refer to as "overlapped genes") (Fig. 3d), we found that the extent of splicing defect is more severe in the latter (Supplementary Fig. 10). This suggests that binding of FgRbp1 to target mRNAs presumably improves the splicing efficiency of its substrate mRNAs. Therefore, FgRbp1 may recruit FgU2AF23 to the 3′-splice site of its target gene by directly binding to its target mRNA, thereby promoting the early assembly of spliceosome and contributing to elevated splicing efficiency of target mRNAs. This is similar to the splicing regulation mechanism in mammals

where some non-snRNP splicing regulators (such as SR protein, YB-1, SAM68) can promote exon inclusion or intron splicing by recruiting U1 snRNP to the 5' splice site or U2AF to the 3' splice site in early steps of spliceosome assembly[25–27].

Splicing errors resulting from either defects of key splicing factors or mutations of splicing regulatory sequences lead to aberrant splicing of multiple transcripts and contribute to various diseases in humans[11]. Similarly, ΔFgRbp1 mutant exhibited various defects in hyphal growth, sexual development, and pathogenicity. The pathogenicity is meticulously regulated by various pathways, and in *F. graminearum* the Gpmk1 MAPK signaling pathway is involved in regulating pathogenicity via controlling the formation of penetration structure and secretion of plant cell wall degrading enzymes[33,34,49]. In this pathway, the deletion mutants of the upstream sensor FgSho1, the MAPK module FgSte50-Ste11-Ste7-Gpmk1, and the downstream transcription factor FgSte12 are unable to penetrate the cellophane and plant[34,35]. Significantly, all these genes exhibited splicing defects in ΔFgRbp1 while the transcription levels of these genes did not alter (Supplementary Data 3c). In addition, it is worth noting that FgRbp1 influences splicing of DON biosynthesis genes (*TRI1, TRI4, TRI5, TRI10*), thus significantly reducing DON levels in ΔFgRbp1 compared with that in the wild type (Supplementary Fig. 11). Therefore, we conclude that the collective splicing defects in pathogenicity related genes contribute to the loss of virulence in ΔFgRbp1 on host plant.

## Methods

**Fungal strains and culture conditions**. *Fusarium graminearum* strain PH-1 (NRRL 31084) was used as the wild-type strain for constructing gene-deletion mutants. Fungal strains were cultured at 25 °C on potato dextrose agar (PDA), complete medium (CM) or minimal medium (MM) for mycelial growth tests. The carboxymethyl cellulose (CMC) liquid medium was used for conidiation assays[50]. For trichothecene production analysis, each strain was grown in liquid trichothecene biosynthesis inducing (TBI) medium[51]. For RNA and protein extraction, vegetative hyphae were harvested from yeast extract peptone dextrose (YEPD, 1% yeast extract, 2% peptone, 2% glucose) liquid medium. Each experiment was repeated three times.

**Construction of gene deletion and complementation mutants**. Mutant alleles for targeted gene deletion were generated using the double-joint (DJ) PCR method[52]. In brief, the 5' and 3' flanking regions of each gene were amplified using primers listed in Supplementary Data 5 and the flanking sequences were then fused with the hygromycin-resistance gene cassette (HPH). The transformation of *F. graminearum* was carried out via polyethyleneglycol (PEG) mediated protoplast transformation method[53]. To obtain *F. graminearum* protoplasts, the fresh mycelia were treated with 30 mg driselase (D9515, Sigma, St.Louis, MO), 200 mg lysozyme (RM1027, RYON, Shanghai, China), and 200 mg cellulose (RM1030, RYON, Shanghai, China). Putative gene-deletion mutants were identified by PCR assays with relevant primers (Supplementary Data 5) and further confirmed by Southern hybridization. For complementation, vectors containing target ORF fused with a tag along with geneticin-resistance gene were transformed into corresponding deletion mutant. To create the FgRbp1-GFP fusion construct, the open reading frame (ORF) of FgRbp1 was amplified using primers listed in Supplementary Data 5. The resulting PCR products were co-transformed with Xho1-digested pYF11 into the yeast strain XK1-25[54]. Subsequently, the FgRbp1-GFP fusion vector was recovered from the yeast transformants and then transferred into *E.coli* strain DH5α for amplification. Other GFP-, RFP- and Flag-fusion constructs were constructed using the same strategy. For RBM42-GFP construction, the coding sequence of human RBM42 (1350 bp) was amplified from the cDNA prepared from HeLa cells according to a previous report[40], and then cloned into pYF11 using the same strategy. All the primers were listed in Supplementary Data 5.

**Yeast two-hybrid screening and verification**. To screen for FgRbp1-interacting proteins, the full-length cDNA sequence of FgRbp1 was cloned into the yeast vector pGBKT7 (Clontech) as a bait. The *F. graminearum* cDNA library was constructed in the Y2H vector pGADT7 using total RNA extracted from mycelia and conidia. The vector pGBKT7-FgRbp1 and cDNA library were co-transformed into the yeast strain Y2H Gold. The resulting transformants were selected using the SD-Trp-Leu-His selective medium. Surviving clones were transferred to SD-Trp-Leu-His-Ade selective medium. Positive clones were sequenced. For the verification assay, the coding sequence of FgU2AF23 was amplified from cDNA of *F. graminearum* and inserted into the pGADT7 vector (prey). The prey and bait (pGBKT7-

FgRbp1) were co-transformed into Y2H Gold strain. Transformants were selected from SD-Trp-Leu medium, gradient-diluted, and plated on SD-Trp-Leu-His-Ade medium at 30 °C for 4 days. Three independent replicates were performed to confirm for each yeast two-hybrid assay.

**Western blotting**. Each tested strain was incubated in YEPD liquid medium with agitation (180 rpm) for 12 h at 25 °C. The fresh mycelia (200 mg) were ground into fine powder in liquid nitrogen, and suspended with 1 mL extraction buffer (50 mM Tris-HCl [pH 7.5], 150 mM NaCl, 5 mM EDTA, 1% Triton X-100, 1:100 v/v protease inhibitor cocktail [Sangon Biotech, Shanghai, China]). After homogenization with a vortex shaker, the lysate was centrifuged at $15000 \times g$ for 20 min at 4 °C. The resulting proteins were separated by 10% sodium dodecyl sulfate-polyacrylamide gel electrophoresis (SDS-PAGE) and transferred to Immobilon-P transfer membrane (Millipore, Billerica, MA, USA)[55]. GFP- and RFP-tagged proteins were detected with monoclonal anti-GFP (ab32146, Abcam, Cambridge, UK, 1:10000 dilution) and anti-RFP (ab65856, Abcam, Cambridge, UK, 1:10000 dilution) antibodies, respectively. The endogenous RPS3 and RPS20 were detected using the anti-RPS3 antibody (ET1601-27, HuaAn Biotechnology, Hangzhou, China, 1:1000 dilution) and anti-RPS20 antibody (ET 1610-83, HuaAn Biotechnology, Hangzhou, China, 1:1000 dilution), respectively. The samples were also detected with monoclonal anti-H3 antibody (ab8895, Abcam, Cambridge, UK, 1:10000 dilution) or anti-GAPDH antibody (EM1101, HuaAn Biotechnology, Hangzhou, China, 1:5000 dilution) as a reference. The intensity of immunoblot bands were quantified using the Image J software. Each experiment was repeated three times.

**Co-immunoprecipitation (Co-IP) assays**. The FgRbp1-GFP and FgU2AF23-RFP fusion constructs were generated as described in our previous section. Both constructs were verified by DNA sequencing and co-transformed into PH-1. Transformants expressing pairs of fusion constructs were confirmed by western blotting. In addition, the transformants bearing a single fusion construct were used as references. For Co-IP assays, total proteins were extracted and incubated with the GFP-Trap Agarose (ChromoTek, Martinsried, Germany) at 4 °C for 6 h with rotation. After six times of washing, proteins eluted from agarose were analyzed by western blot detection with anti-RFP (ab65856, Abcam, Cambridge, UK, 1:10000 dilution) antibody. The input samples were also detected with monoclonal anti-GAPDH antibody described in the previous section as a reference. To detect whether the interaction between FgRbp1 and FgU2AF23 depended on RNA, the cell lysate was treated with 500 μg/ml RNase A at room temperature for 30 min prior to immunoprecipitation[56]. The experiment was repeated twice.

**Bimolecular fluorescence complementation (BiFC) assays**. To produce constructs for BiFC, the full-length *FgRBP1* was amplified and cloned into pHZ65 vector that carries NYFP and the hygromycin-resistance marker to generate the FgRbp1-NYFP fusion construct as described[57]. Using the same approach, the FgU2AF23-CYFP fusion construct was generated by cloning the full-length *FgU2AF23* into pHZ68 vector that carries CYFP and the zeocin-resistance marker. Both FgRbp1-NYFP and FgU2AF23-CYFP constructs were verified by sequencing and then transformed in pair into PH-1::H1-RFP strain (a strain that expressed RFP-tagged histone 1 protein in PH-1 to visualize nuclei). Construct pairs of FgRbp1-NYFP and CYFP, FgU2AF23-CYPF and NYFP were also co-transformed into PH-1::H1-RFP strain to serve as negative controls. Transformants resistant to both hygromycin and neomycin were isolated and confirmed by PCR. YFP signals in the mycelia were examined under a Zeiss LSM780 confocal microscope (Gottingen, Niedersachsen, Germany).

**RNA-seq analysis**. To harvest fresh mycelial samples for RNA-seq, 1 ml of conidial suspension ($10^6$ conidia/ml) of each strain was inoculated and incubated in 200 ml YEPD liquid medium with agitation (180 rpm) for 12 h at 25 °C. Total RNA was isolated from the harvested mycelia with the TRIzol Reagent (Life technologies, US) and shipped to Sinotech Genomics Corporation (Shanghai, China) for library construction and sequencing with Illumina Hiseq X-Ten sequencer. Three biological replicates for each strain were sequenced. For each sample, 35–50 M clean RNA-seq reads were obtained and mapped to the *F. graminearum* PH-1 genome sequence using Hisat2. For comparison and visualization of splicing efficiency, intron retention rate was determined by calculating the ratio of introns FPKM (Fragments Per Kilobase Million) to the gene's overall exon FPKM for each tested intron[19]. To filter out weakly expressed genes, only genes with a minimum expression level of 1 count per million were included according to the previous RNA-seq analysis in *F. graminearum*[17]. The intron retention rate was filtered for an adjusted $p$ value $< 0.05$ whether in the wild type or mutant. After completing the above filtering, genes with intron retention rates greater than twice that of wild type were defined as intron retention genes in ΔFgRbp1.

**RIP-seq and RIP-qPCR analysis**. RIP-seq was performed as described previously with minor modifications[58–60]. Briefly, RIP was performed using the complementary strain expressing FLAG-tagged FgRbp1 in ΔFgRbp1 (ΔFgRbp1:: FgRbp1-FLAG). Fresh mycelia samples were harvested as described earlier in RNA-seq methodology. Fresh mycelia were ground into fine powder in liquid

nitrogen and suspended with 4 mL lysis buffer (50 mM Tris-HCl [pH 7.5], 150 mM KCl, 2 mM EDTA, 0.5% NP-40, 0.5 mM dithiothreitol, 1:100 v/v protease inhibitor cocktail [Sangon Biotech, Shanghai, China], 200 units/ml RNase OUT [Invitrogen]), followed by simple sonication (30 s on, 30 s off, repeated twice). Homogenates were centrifuged for 20 min at 16000 × g under 4 °C to clear the lysate, and a 200-µl sample was saved as the input sample. The remaining sample was divided into two equal parts (IP sample and mock sample) and incubated with anti-Flag M2 (Sigma, F1804) antibody (IP sample antibody) or anti-IgG1 (Thermo Scientific, MA1-10406) antibody (mock sample antibody) together with the magnetic protein G beads (Life Technologies; 10004D) at 4 °C overnight with rotation. After incubation, the bead-protein-RNA complexes were washed with ice-cold NT2 buffer (50 mM Tris-HCl (pH 7.4), 1 mM MgCl$_2$, 150 mM NaCl, 0.05% NP40, 0.5 mM dithiothreitol, 1:100 v/v protease inhibitor cocktail and 200 units/ml RNase OUT) for five times. After the last washing step, the beads were treated with 30 µg proteinase K at 55 °C for 30 min to release the RNP complexes and RNA was then extracted with Trizol reagent. IP and input RNAs were shipped to Novogene for library construction and sequencing on Illumina HiSeq 2500 (Novogene, Beijing, China) or used for quantitative PCR analysis.

For RIP-qPCR, RNAs from input, IP and mock sample were reverse transcribed into cDNA and subjected to qRT-PCR using SYBR green I fluorescent dye detection. Primer sequences are described in Supplementary Data 5. Fold enrichment of tested genes was calculated using the equation $2^{\Delta(Ct(mock-input) - Ct(IP-input))}$. Meanwhile, ACTIN was also amplified as a negative control. For every analyzed RNA fragment, each sample was quantified from three independent RIP analyses.

**Electrophoretic mobility shift assay.** In order to prepare GST-fused FgRbp1 or FgRbp1$^{\Delta RRM}$, coding sequences of FgRbp1 or FgRbp1$^{\Delta RRM}$ were cloned into pGEX4T-3 (GE Healthcare Life Science), after which the resulting constructs were transformed into Escherichia coli BL21 cells for protein expression. BL21 cells expressing GST-fused proteins were grown in 600 ml LB liquid medium at 37 °C until the OD600 reached 0.6, followed by induction of expression at 16 °C for 6 h in the presence of 1 mM isopropyl β-D-1-thiogalactopyranoside. After lysing cells, the recombinant proteins were purified on glutathione sepharose beads (Sangon Biotech, Shanghai, China) and eluted with 5 mM glutathione dissolved in TBS buffer. RNAs of target genes containing the peaks' sequences (~200–300 bp in length) were achieved by in vitro transcription using T7 RNA polymerase following the manufacturer's instructions (Takara). The sequence of six repeats of 5'-CAAGA-3', 5'-CAAGG-3', 5'-ACGAA-3' and 5'-GUUCU-3'was synthesized from Sangon Biotech (Shanghai, China).

For the binding reaction, 1 µg in vitro transcribed mRNA or 2 µM synthesized (5'-CAAGR-3')$^6$ RNA was incubated with 5 µM different recombinant proteins in 20 µL of 1 x buffer containing 10 mM Tris-HCl (pH 7.5), 20 mM KCl, 1 mM MgCl$_2$, and 1 mM DTT at room temperature for 5 min. After incubation, the samples were separated on 2.0% agarose gel for 30 min at 150 V at room temperature using 0.5 x TBE. After electrophoresis, gels were stained for 40 min with gentle shaking in 50 mL SYBR Gold (Invitrogen) containing buffer, which was created by diluting 10 µl SYBR Gold stock into 50 mL 0.5 x TBE buffer[61]. Finally, agarose gels were imaged using a Bio-Rad Molecular Imager.

**Measurement of splicing ratio.** Total RNAs were extracted from fresh mycelia and treated with DNAse to remove residual DNA before reverse transcription. Quantitative real-time PCR was performed to measure RNA levels with three biological repeats. Measurement of splicing ratio was conducted using the previously described protocol[19,23,24]. In brief, the splicing ratio was calculated by determining the level of spliced RNA normalized to the level of unspliced RNA for each intron tested using the equation $2^{\Delta(Ct-unspliced - Ct-spliced)}$. The splicing ratio of wild-type strain was normalized to 1. For amplifying the spliced RNA, the forward primer was designed to cross exon-exon junctions to ensure specific amplification of spliced RNA (Supplementary Fig. 12), and the reverse primer was designed targeting the exon adjacent to the intron (Supplementary Fig. 12). For amplifying the unspliced RNA, the forward primer was designed within the intron (Supplementary Fig. 12), and the reverse primer was the same as that for amplifying the spliced RNA (Supplementary Fig. 12). Both spliced and unspliced primers were listed in Supplementary Data 5. The experiment was repeated three times independently.

**RNA affinity selection assay.** The biotin-labeled FgRPS20 mRNA fragment containing exon1-intron1-exon2 (as shown in Fig. 5f) was achieved by in vitro transcription with T7 polymerase (Takara) and Biotin RNA Labeling Mix (Roche). Streptavidin magnetic beads (NEB, S1420S) were washed with buffer (0.5 M NaCl, 20 mM Tris-HCl, pH 7.5, 1 mM EDTA) for three times. Three microgram of biotinylated FgRPS20 mRNA fragment was immobilized on streptavidin magnetic beads by incubation with 20 µl beads at 4 °C for 4 h. After incubation, the beads were washed three times with washing buffer (20 mM HEPES, pH 7.5, 1 mM DTT, 10 mM MgCl$_2$, 300 mM NaCl, 0.01% NP-40, RNase inhibitor). After washing, beads immobilized with biotin-labeled RNA were incubated with purified single protein: GST-FgRbp1, GST-FgRbp1$^{\Delta N}$, FgU2AF23-His, or pairs of proteins: GST-FgRbp1/ FgU2AF23-His, GST-FgRbp1$^{\Delta N}$/FgU2AF23-His in binding buffer (20 mM HEPES, pH 7.5, 1 mM DTT, 10 mM MgCl$_2$, 150 mM NaCl, RNase inhibitor)

at 4 °C for 1.5 h with rotation. Beads were washed with previously mentioned washing buffer containing 300 mM NaCl for 5 times and the bound proteins were detected by western blot with anti-GST antibody (EM80701, HuaAn Biotechnology, Hangzhou, China, 1:10000 dilution) or anti-His antibody (ab18184, Abcam, Cambridge, UK, 1:5000 dilution).

**Pathogenicity and DON biosynthesis assays.** To assess pathogenicity of each strain on flowering wheat heads, a 10 µl aliquot of conidial suspension ($10^5$ conidia ml$^{-1}$) was injected into a floret in the central section spikelet of single flowering wheat heads of susceptible cultivar Jimai 22. There were 15 replicates for each strain. Fifteen days after inoculation, the infected spikelets in each inoculated wheat head were recorded. To evaluate pathogenicity on corn silks, fresh corn silks were placed on sterilized filter paper soaked with sterile water. The central of the corn silks were inoculated with a 5-mm-diameter fresh mycelial plug. Infection of corn silk was measured based on the extent of discoloration after 5 days of incubation at 25 °C[62]. All experiments were repeated three times. To analyze the virulence of the FgRbp1 mutants in detail, hyphal structures from infected wheat spikelets at 7 d post inoculation (dpi) were examined with a Hitachi TM-1000 tabletop microscope (Hitachi, Tokyo, Japan). The penetration behavior of each strain was examined on cellophane membranes[63]. Briefly, each strain was inoculated onto the cellophane-covered MM medium. After incubation for 3 days at 25 °C, the cellophane together with the colonies of each strain on the cellophane were removed. Then, the plates were incubated for three additional days to examine for hyphal growth. The experiment was repeated three times. To determine the DON production, each strain was grown in TBI liquid medium at 28 °C for 7 days in a shaker (150 rpm) in the dark. Then, the DON production for each sample was extracted and quantified by using a DON Quantification Kit Wis008 (Wise Science, Zhenjiang, China).

**Statistics and reproducibility.** Representative images from three independent experiments are presented in Figs. 2a, d–f, 4b, 7b, d, and Supplementary figures 5, 7. Electrophoretic mobility shift assays were carried out two times and representative images are shown in Figs. 4c, 5e, g, 6d, 7g. RNA affinity selection assay was performed three times, and representative images are shown in Fig. 5h. PCR identification and southern blotting were carried out two times and representative images are presented in Supplementary figures 2b and c.

**Reporting summary.** Further information on research design is available in the Nature Research Reporting Summary linked to this article.

## Data availability
Data supporting the major findings of this work are available within the paper and its Supplementary Information files. The RNA-seq data, RIP-seq data has been deposited in the NCBI SRA database with accession codes PRJNA556958 and PRJNA556748, respectively. All relevant data are available from the corresponding authors upon any reasonable request. Source data are provided with this paper.

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

## Acknowledgements

We thank Dr. Yunpeng Gai for bioinformatics analysis of intron retention rate. This work was financially supported by the National Key R&D Program of China (2018YFE0206000), the National Natural Science Foundation of China (31930088, 31922074), China Agriculture Research System (CARS-3-1-29).

## Author contributions

Z.M., Y.C., and M.W. designed the experiments. M.W., T.M., and H.W. performed the experiments and analyzed the data. J.L. provided technical support and discussions. Z.M., Y.C. W.B.S., and M.W. wrote the paper.

## Competing interests

The authors declare no competing interests.
