## [Peer Review File · Nature Communications]

Reviewers' comments:

Reviewer #1 (Remarks to the Author):

In this study, Wang and colleagues have identified a new RNA binding protein in *Fusarium graminearum* whose deletion results in dramatic splicing defect characterized essentially by intron retention.

In *Fusarium graminearum*, through the systematic deletion of uncharacterized RRM containing proteins, the authors have identified FgRbp1 as an essential growth factor. Complementation with FgRbp1's human ortholog RBM42 restored growth in Δ FgRbp1 *Fusarium* strains indicating that FgRbp1 is functionally conserved. Further analysis show that FgRbp1's RRM is essential for FgRbp1 function as its deletion results in severe growth defect reminiscent of the one observed in Δ FgRbp1 strains.

Y2H experiments have revealed that, amongst other interactors, FgRbp1 interacts with the splicing factor U2AF23. BiFC experiments indicate that FgRbp1-U2AF23 interaction is restricted to the nucleus suggesting a potential role in splicing.

This was further substantiated by RNA-seq experiments: the deletion of FgRbp1 resulted in massive intron retention i.e. 8780 retained introns in 4933 transcripts.

This effect on splicing seems, at least partially, mediated through the direct binding of FgRbp1 with its target pre-mRNA. Indeed RIP-seq experiments identified 219 pre-mRNAs that would be directly bound by FgRbp1 through a CAAGR cis element.

From the 219 FgRbp1 pre-mRNA targets identified by RIP-seq, 165 have their splicing affected upon FgRbp1 deletion. Interestingly, from those 165 mRNAs, 68 are coding for ribosomal proteins. Accordingly, further analysis show that FgRPS20 is spliced in a FgRbp1 and -CAAGR- dependent manner.

Similarly, FgRbp1 regulates the splicing of essential sexual reproduction genes. The authors show that upon deletion of FgRbp1 the splicing of FgMCM1 and FgGPA1 is compromised.

Finally, the author demonstrate that *Fusarium graminearum*'s virulence is strongly attenuated as a result of FgRbp1 deletion. This effect on virulence would be conveyed by FgSte12 mis-splicing.

This manuscript is clearly written and experiments are in general well performed. Although the data presented in this study are novel they may not represent a significant step in the understanding of splicing regulation in *Fusarium graminearum*.

1- The author have dedicated a lot of effort in characterizing the interaction between U2AF23 and FgRbp1. As presented in this manuscript, this interaction is the only mechanistic element provided in order to understand the role of FgRbp1 in splicing regulation. While this interaction is clearly established, the authors do not demonstrate whether this interaction is relevant for explaining the role of FgRbp1 in splicing. What would happen if the interaction between U2AF23 and FgRbp1 would be abolished (e.g. by truncation or mutation) while these two factor would still associate with their pre-mRNA target? Would this recapitulate the effect on splicing observed upon FgRbp1 deletion? Considering the data presented in Table S2, FgRbp1 interacts with several other core splicing factors. Why have the authors chosen to focus on the interaction between U2AF23 and FgRbp1 exclusively if the relevance of this interaction is not, even from far, investigated? As presented here, the characterization of the U2AF23-FgRbp1 interaction does not bring much to the overall story. The significance of this manuscript would greatly beneficiate from a functional characterization of this interaction i.e. what is the impact on splicing when U2AF23 and FgRbp1 cannot interact with each other.

2- RBM42 should be presented in the introduction. It comes out of the blue in the result section. Moreover, the authors show that RBM42 can rescue FgRbp1 deletion although the sequence identity is only restricted to the RRM motif. Indeed, the regions flanking the RRM are clearly divergent between FgRbp1 and RBM42. This raises the question whether RBM42 can interact with U2AF23. If this interaction (RBM42-U2AF23) would be retained, this would be a good point in favor of the functional importance of the interaction between FgRbp1 and U2AF23. Inversely, if RBM42 would not interact with U2AF23 this would be an interesting finding suggesting that the growth defect resulting from FgRbp1 deletion would occur independently of its ability to interact with U2AF. Therefore I would strongly encourage the authors to investigate whether RBM42 interacts with U2AF23.

3- The splicing of 8780 introns located in 4933 transcripts is compromised upon FgRbp1 deletion. However, RIP-seq identifies only 219 mRNA that are bound by FgRbp1.

These two experiments indicate that (i) deletion of FgRbp1 has a broad/generic effect on splicing (ii) FgRbp1 binds only a very limited subset of transcripts. How the authors would explain those apparently conflicting results?

Moreover, this raises one fundamental question: is the broad effect of FgRbp1 deletion on splicing a direct/primary effect or rather a secondary effect resulting from the splicing defect of the few transcripts shown to interact with FgRbp1. Indeed, as convincingly shown, FgRbp1 directly binds and regulate the splicing of 68 transcripts coding for ribosomal proteins. The down-regulation of those ribosomal proteins resulting from FgRbp1 deletion has, very likely, strong deleterious effect on the cell's translational capacity. This, as a consequence, should have a broad effect on the overall cell metabolism. This point is not approached by the authors neither experimentally nor conceptually. This should be, at the very least, extensively addressed in the discussion.

4- The authors show that FgRbp1 binds preferentially/specifically to the CAAGR cis-element. EMSA experiments clearly indicate that FgRbp1 binds this sequence efficiently. However a specificity control is missing. How would FgRbp1 bind to a scrambled sequence e.g. RACGA or an unrelated sequence of the same length like CCCCC, AAAAA etc...? This point should be definitely addressed.

5- Fig. 4 panel I: does the mutated FgRPS20 construct, without the CCAAGAACCTCAAGA sequence, interact with FgRbp1? It is an important control. Indeed the mutated FgRPS20 construct may still interact with FgRbp1.

Moreover, line 271, the authors conclude "These results indicate that FgRbp1 regulates the splicing of FgRPS20 intron 1 through direct interaction with the cis-element CAAGR." Although all experiments presented in Fig 4 suggest/support this conclusion, this is, however, not demonstrated. As mentioned above, this point could be made only if the authors would test by EMSA whether FgRPS20-mu interacts with FgRbp1.

6- New experiments and results, not shown in the result section, are presented and detailed in the discussion section between the line 364 and 400!!! Moreover, the way those data are presented is highly speculative (this is certainly why they are presented in the discussion). This part should be either moved to the result section or deleted.

Results shown in Table S2 are only mentioned in the discussion part. It should be detailed in the result section.

In general, the Discussion section should be deeply changed...it should be a discussion not a result section.

7- The author do not investigate nor provide a potential mechanistic model how FgRbp1 would regulate splicing. The lack of mechanistic insights strongly hinders the impact of this study.

Line 225, "RPGs" should be defined.

Line 250, "PGRs" should be spelled RPGs.

Line 278, the author should define " Δ FgRbp1-C".

Are the FgRPS20 mRNA used in Fig. 4 panel I and H similar? This point does not clearly appear in the text or figure legend.

Do FgMCM1 and FgGPA1 transcripts contain the CAAGR motif? This should be explicitly mentioned.

Line 365, Table S3 and not S2 should be referenced.

Reviewer #2 (Remarks to the Author):

The manuscript by Wang et al. describes the function of a conserved RRM domain RNA binding protein in a filamentous fungus. This RNA binding protein, FgRbp1, regulates splicing efficiency of about half of the *F. graminearum* introns and binds to a conserved splicing factor. Significantly, it can be functionally replaced by its human ortholog, whose function has not been described in detail yet. The authors performed RNA-seq and RIP-seq to identify transcripts influenced and bound by FgRBP1 and they identified a binding motif that they verified by EMSA studies. They elegantly show that splicing of introns depends on this binding motif (Fig. 4I). Further, they provide evidence that the splicing defect leads to the pleiotropic phenotype of the Δ FgRbp1 mutant, by showing splicing defects in transcripts for ribosomal proteins, sexual developmental genes, and virulence factors. Overall, this manuscript covers a highly interesting, highly conserved topic and provides evidence for distinct functions of a conserved protein of the eukaryotic splicing machinery. Although the conclusions drawn from the results seem justified, I have major concerns about the presentation and clarity of the data. I would therefore suggest careful rewriting of the manuscript.

The manuscript contains many data that have been organized into 6 figures and 9 supplementary figures as well as supplementary tables and datasets. Several supplemental figures describing results are only mentioned in the discussion, although they represent results that have not been covered in the results section. For instance, the part on the SAGA complex represents results; however, to me these results on *gcn5* seem much more preliminary than the results on FgRbp1 and the conclusions therefore speculative. This part should be either clarified and put in the results section or it should be omitted from the manuscript. Due to the SAGA part in the discussion, the discussion in total is rather short.

The authors estimate the intron retention rate by comparing reads mapping to intronic versus exonic regions. However, this approach does not take into account sequencing bias. I would suggest using a more substantial approach for estimating splicing efficiency, e.g. calculating PIR (PIR was calculated as the percentage of the average number of reads mapping to the 5' and 3' exon-intron junctions, over the sum of the average of the exon-intron junction reads plus the exon-exon junction reads; from Braunschweig et al. 2014 Genome Res 24:1774-1786)

It is often unclear to me, especially in first reading, how an analysis was performed and how the controls were done, how many introns a transcript contains and how many were tested (this information is also often lacking in the schematics provided in the figures).

I have a number of additional concerns, listed, but not limited to my further comments:

- Strain designation Δ FgRbp1-C: Maybe I missed it, but what strain is this?
- Does the Rbp1-FLAG construct used for affinity capture complement the mutant phenotype?
- Fig. 2: What is positive and negative control in Fig. 2C? Where is the control for the pGADT7-FgU2AF23 construct? What dilutions were plated? Similar for further yeast two hybrid analysis in this

manuscript

- Fig. 2E: Please explain how you can distinguish YFP and RFP with an epifluorescence microscope?
- Please describe in more detail how distinct transcripts were tested for splicing efficiency. Line 577, "spliced and unspliced primers", is misleading.
-
- For RNA-seq and RIP-seq, were the same growth conditions used? Please define "fresh mycelia" in line 520.
- Which genome version were the sequencing data mapped to? Which genome version do the Locus tag numbers refer to?
- Information on how the phylogenetic analysis was performed is missing. Is the tree in Fig. S3 based on the alignment in Fig. S4? If so, which part?
- Data availability statement is missing
- Line 569, define "fresh mycelia"
- Line 233, "induced by these RPGs", should this read "induced IN these RPGs"?
- Line 319, "both the first and second intron of FgSTE12, reads like there are more than two introns, but legend to Fig. 6 implies there are only two.
- I would suggest putting Fig. S7A together with Fig. 4D, since this is one central result of the manuscript
- The legends need to be revised to be more clear about what is shown and what has been done. For example, legend to Fig. S7A is much clearer than legend to Fig. 4D, which I think is wrong (this is a qRT-PCR analysis of the splicing, not the transcript level)
- When reading the manuscript, I am confused reading about conidiation assays in the Methods section (line 424), but not finding conidiation assays in the results. There are several more examples where methods and results description or legends do not match. For example, the legend to Fig. 2 mentions epifluorescence microscopy, while the methods section describes confocal microscopy (which would also make more sense)
- Fig. S2, labeling in C not consistent with legend. Figure needs more explanation / schematic representation needs more information.
- Error bars are not always defined in the legends.
- Line 835-836, "inoculation... at 25 °C", this does not make sense
- The manuscript should be proofread by a native speaker, since there are a number of small mistakes

Response to Reviewer's comments and suggestions

Reviewer #1 (Remarks to the Author):

In this study, Wang and colleagues have identified a new RNA binding protein in *Fusarium graminearum* whose deletion results in dramatic splicing defect characterized essentially by intron retention. In *Fusarium graminearum*, through the systematic deletion of uncharacterized RRM containing proteins, the authors have identified FgRbp1 as an essential growth factor. Complementation with FgRbp1's human ortholog RBM42 restored growth in Δ FgRbp1 *Fusarium* strains indicating that FgRbp1 is functionally conserved. Further analysis show that FgRbp1's RRM is essential for FgRbp1 function as its deletion results in severe growth defect reminiscent of the one observed in Δ FgRbp1 strains. Y2H experiments have revealed that, amongst other interactors, FgRbp1 interacts with the splicing factor U2AF23. BiFC experiments indicate that FgRbp1-U2AF23 interaction is restricted to the nucleus suggesting a potential role in splicing. This was further substantiated by RNA-seq experiments: the deletion of FgRbp1 resulted in massive intron retention i.e. 8780 retained introns in 4933 transcripts. This effect on splicing seems, at least partially, mediated through the direct binding of FgRbp1 with its target pre-mRNA. Indeed RIP-seq experiments identified 219 pre-mRNAs that would be directly bound by FgRbp1 through a CAAGR cis element. From the 219 FgRbp1 pre-mRNA targets identified by RIP-seq, 165 have their splicing affected upon FgRbp1 deletion. Interestingly, from those 165 mRNAs, 68 are coding for ribosomal proteins. Accordingly, further analysis show that FgRPS20 is spliced in a FgRbp1 and – CAAGR- dependent manner. Similarly, FgRbp1 regulates the splicing of essential sexual reproduction genes. The authors show that upon deletion of FgRbp1 the splicing of FgMCM1 and FgGPA1 is compromised. Finally, the author demonstrate that *Fusarium graminearum*'s virulence is strongly attenuated as a result of FgRbp1 deletion. This effect on virulence would be conveyed by FgSte12 mis-splicing.

This manuscript is clearly written and experiments are in general well performed. Although the data presented in this study are novel they may not represent a significant step in the understanding of splicing regulation in *Fusarium graminearum*.

Comment 1: The author have dedicated a lot of effort in characterizing the interaction between U2AF23 and FgRbp1. As presented in this manuscript, this

interaction is the only mechanistic element provided in order to understand the role of FgRbp1 in splicing regulation. While this interaction is clearly established, the authors do not demonstrate whether this interaction is relevant for explaining the role of FgRbp1 in splicing. What would happen if the interaction between U2AF23 and FgRbp1 would be abolished (e.g. by truncation or mutation) while these two factor would still associate with their pre-mRNA target? Would this recapitulate the effect on splicing observed upon FgRbp1 deletion?

Re: We thank the reviewer for valuable comments. As suggested, we designed a series truncated FgRbp1 constructs and found that the N-terminal domain in FgRbp1 is essential for interaction with FgU2AF23 (Fig 2B). The deletion of N-terminal domain also led to the intron splicing defect, although the extent of defect was not as serious as that observed in Δ FgRbp1 (Fig 5A). In addition, the N-terminal domain deletion strain exhibited obstructed growth and decreased virulence (Fig 2C and Fig 7A). These results indicate that the interaction of FgRbp1-FgU2AF23 is responsible for the role of FgRbp1 in splicing, and has certain influence on the growth and pathogenicity of the fungus.

The study from human cells on splicing regulation mechanism has shown that many splicing regulators (e.g. SR protein or other RBPs) can promote exon inclusion or intron splicing by recruiting U1 snRNP to the 5' splice site or U2 auxiliary factor (U2AF) to the 3' splice site through protein-protein interactions in early steps of spliceosome assembly^{1,2,3}. Since U2AF35 in human (the ortholog of FgU2AF23) is an essential protein in the first step of assembly of spliceosome, we presumed that FgRbp1-FgU2AF23 interaction can also promote the recruitment of FgU2AF23 to FgRbp1-bound mRNAs to regulate intron splicing. To test this, *in vitro* RNA affinity selection assay was conducted using *FgRPS20* mRNA as a representative example. The biotin-labeled *FgRPS20* mRNA fragment containing exon1-intron1-exon2 (as shown in Fig 5F) was *in vitro* transcribed and immobilized on streptavidin magnetic beads. The baits were incubated with either *in vitro* purified GST-FgRbp1 or FgU2AF23-His (Fig S7), or both proteins; while the interaction domain deleted protein GST-FgRbp1 ^{Δ N} (Fig S7) was used as a control. As observed in Fig 5H, GST-FgRbp1 alone bound to the *FgRPS20* RNA baits, which is consistent with the EMSA result in Fig 5. Meanwhile FgU2AF23-his alone also bound to the RNA baits, which is in line with the previous report for human U2AF35 or yeast U2AF23 binds to AG dinucleotide at the 3' splice site in intron^{4,5}. However, when the two proteins

were mixed together and incubated with the RNA baits, the binding of FgU2AF23-his to the RNA was significantly increased while the binding ability of FgRbp1 did not alter (Fig 5H). Moreover, GST-FgRbp1^{ΔN} could also alone bind to the *FgRPS20* RNA baits, suggesting that RRM domain is the determinant of FgRbp1's binding ability. But the addition of GST-FgRbp1^{ΔN} did not increase the FgU2AF23-his binding to the RNA baits (Fig 5H), indicating that this recruitment is dependent on the interaction between the two proteins. These results demonstrated that FgRbp1 can enhance the recruitment of FgU2AF23 to the target *FgRPS20* mRNA via interacting with FgU2AF23.

Comment 2: Considering the data presented in Table S2, FgRbp1 interacts with several other core splicing factors. Why have the authors chosen to focus on the interaction between U2AF23 and FgRbp1 exclusively if the relevance of this interaction is not, even from far, investigated? As presented here, the characterization of the U2AF23-FgRbp1 interaction does not bring much to the overall story. The significance of this manuscript would greatly benefit from a functional characterization of this interaction i.e. what is the impact on splicing when U2AF23 and FgRbp1 cannot interact with each other.

Re: In our original manuscript, we described that U2AF23 is the only protein involved in mRNA splicing identified in Y2H screening assay. To identify other potential interacting proteins involved in splicing, we conducted affinity capture-mass spectrometry assay, and captured eight other splicing-associated proteins. The interaction between FgRbp1 and these other eight splicing factors may be indirect as affinity capture assays tend to pull down numerous indirect interacting proteins. We performed Y2H assay to verify if the interactions between FgRbp1 and these eight proteins are physically direct. The results showed that none of these eight proteins directly interacted with FgRbp1 in Y2H assay (as shown in the figure below), suggesting that these proteins interact with FgRbp1 indirectly. Since these proteins do not interact with FgRbp1 directly, we delete Table S2 from our original submission in this revision.

Comment 3- RBM42 should be presented in the introduction. It comes out of the blue in the result section. Moreover, the authors show that RBM42 can rescue FgRbp1 deletion although the sequence identity is only restricted to the RRM motif. Indeed, the regions flanking the RRM are clearly divergent between FgRbp1 and RBM42. This raises the question whether RBM42 can interact with U2AF23. If this interaction (RBM42-U2AF23) would be retained, this would be a good point in favor of the functional importance of the interaction between FgRbp1 and U2AF23. Inversely, if RBM42 would not interact with U2AF23 this would be an interesting finding suggesting that the growth defect resulting from FgRbp1 deletion would occur independently of its ability to interact with U2AF. Therefore I would strongly encourage the authors to investigate whether RBM42 interacts with U2AF23.

Re: As you suggested, we tested the interaction between RBM42 and FgU2AF23, and found that RBM42 interacts with FgU2AF23 in Y2H assay (Fig S9). Given that RBM42 can fully restore the growth defect in Δ FgRbp1, the retained interaction between RBM42 and U2AF23 suggest that interaction between FgRbp1 and U2AF23 is functionally important.

Comment 4- The splicing of 8780 introns located in 4933 transcripts is compromised upon FgRbp1 deletion. However, RIP-seq identifies only 219 mRNA that are bound

by FgRbp1. These two experiments indicate that (i) deletion of FgRBp1 has a broad/generic effect on splicing (ii) FgRbp1 binds only a very limited subset of transcripts. How the authors would explain those apparently conflicting results? Moreover, this raises one fundamental question: is the broad effect of FgRbp1 deletion on splicing a direct/primary effect or rather a secondary effect resulting from the splicing defect of the few transcripts shown to interact with FgRbp1. Indeed, as convincingly shown, FgRbp1 directly binds and regulate the splicing of 68 transcripts coding for ribosomal proteins. The down-regulation of those ribosomal proteins resulting from FgRbp1 deletion has, very likely, strong deleterious effect on the cell's translational capacity. This, as a consequence, should have a broad effect on the overall cell metabolism. This point is not approached by the authors neither experimentally nor conceptually. This should be, at the very least, extensively addressed in the discussion.

Re: Thank you very much for your insightful points. We have discussed this issue in the revised manuscript (lines 475-490).

Comment 5- The authors show that FgRbp1 binds preferentially/specifically to the CAAGR cis-element. EMSA experiments clearly indicate that FgRbp1 binds this sequence efficiently. However a specificity control is missing. How would FgRbp1 bind to a scrambled sequence e.g. RACGA or an unrelated sequence of the same length like CCCCC, AAAAA etc...? This point should be definitely addressed.

Re: We have supplemented the specificity control in EMSA experiments according to your suggestion. As shown in Fig 4C, the full-length FgRbp1 could bind to the synthetic RNA sequence of six repeats of 5'-CAAGA-3' or 5'-CAAGG-3', whereas it was unable to bind to a scrambled RNA sequence 5'-ACGAA-3', nor a mutated sequence 5'-GUUCU-3' with six repeats, suggesting that FgRbp1 binds specifically to the CAAGR *cis*-element.

Comment 6- Fig. 4 panel I: does the mutated FgRPS20 construct, without the CCAAGAACCTCAAGA sequence, interact with FgRbp1? It is an important control. Indeed the mutated FgRPS20 construct may still interact with FgRbp1. Moreover, line 271, the authors conclude "These results indicate that FgRbp1 regulates the splicing of FgRPS20 intron 1 through direct interaction with the cis-element CAAGR." Although all experiments presented in Fig 4 suggest/support this conclusion, this is,

however, not demonstrated. As mentioned above, this point could be made only if the authors would test by EMSA whether FgRPS20-mu interacts with FgRbp1.

Re: To investigate whether FgRbp1 can still interact with mutated *FgRPS20* construct *in vitro*, EMSA experiment was carried out by using mRNA fragments of *FgRPS20*-WT and *FgRPS20*-Mu, which were *in vitro* transcribed from the wild-type construct (WT) and the mutated construct (Mu), respectively, as shown in Fig 5F. The sequence of the mRNA fragment *FgRPS20*-WT is identical to that of exon 2 (267-bp) as indicated in Fig 5F, and *FgRPS20*-Mu is consistent to the *FgRPS20*-WT except that the 5'-CCAAGAACCTCAAGA-3' sequence in exon 2 was deleted. The results of EMSA showed that FgRbp1 bound to the mRNA sequence of *FgRPS20*-WT, but hardly to the sequence of *FgRPS20*-Mu (Fig 5G). This suggests that the reduction of splicing efficiency of *FgRPS20* intron 1 (in the strain containing the mutated construct) is due to the loss of binding ability of FgRbp1 to the mutated mRNA.

Comment 7- New experiments and results, not shown in the result section, are presented and detailed in the discussion section between the line 364 and 400. Moreover, the way those data are presented is highly speculative (this is certainly why they are presented in the discussion). This part should be either moved to the result section or deleted. Results shown in Table S2 are only mentioned in the discussion part. It should be detailed in the result section. In general, the Discussion section should be deeply changed...it should be a discussion not a result section.

Re: As stated in the response to comment #2, Y2H assay demonstrated that eight putative splicing-associated proteins identified by affinity capture-mass spectrometry assays do not interact with FgRbp1 directly. The results were presented in the revision (lines 173-176).

Comment 8- The author do not investigate nor provide a potential mechanistic model how FgRbp1 would regulate splicing. The lack of mechanistic insights strongly hinders the impact of this study.

Re: Please see the response to comment #1

Comment 9- Line 225, "RPGs" should be defined.

Comment 10- Line 250, "PGRs" should be spelled RPGs.

Re: We have defined and made a correction for "RPGs" in line 255.

Comment 11- Line 278, the author should define “ Δ FgRbp1-C”.

Re: Δ FgRbp1-C represents the complemented strain Δ FgRbp1::FgRbp1-GFP generated by transformation of Δ FgRbp1 with the full-length FgRbp1 protein fused with GFP. To be consistent with the text in Fig 1 and other contents in the manuscript, we have replaced “ Δ FgRbp1-C” by “ Δ FgRbp1::FgRbp1-GFP” in this revision.

Comment 12- Are the FgRPS20 mRNA used in Fig. 4 panel I and H similar? This point does not clearly appear in the text or figure legend.

Re: The *FgRPS20* mRNA used in the original manuscript Fig. 4 panel I and H is similar. In fact, the sequence of the *in vitro* transcribed mRNA fragment of *FgRPS20* (in the revised manuscript Fig 5E and 5G) is identical to the full-length sequence of exon2 (267-bp) as indicated in Fig 5F in the revised manuscript.

Comment 13- Do FgMCM1 and FgGPA1 transcripts contain the CAAGR motif? This should be explicitly mentioned.

Re: Thanks for raising this concern. *FgGPA1* contains a CAAGA motif in the identified 200-bp peak fragment, while *FgMCM1* contains no CAAGR motif in the identified 173-bp peak fragment (Dataset S3A). Although the gene *FgMCM1* does not contain the CAAGR motif in the peak fragment of RIP, this peak fragment bound by FgRbp1 still contributes to splicing. We also discussed this issue in the revised manuscript (lines 491-511).

Comment 14- Line 365, Table S3 and not S2 should be referenced.

Re: We revised the manuscript as suggested.

Reviewer #2 (Remarks to the Author):

The manuscript by Wang et al. describes the function of a conserved RRM domain RNA binding protein in a filamentous fungus. This RNA binding protein, FgRbp1, regulates splicing efficiency of about half of the *F. graminearum* introns and binds to a conserved splicing factor. Significantly, it can be functionally replaced by its human ortholog, whose function has not been described in detail yet. The authors performed RNA-seq and RIP-seq to identify transcripts influenced and bound by FgRBP1 and they identified a binding motif that they verified by EMSA studies. They elegantly

show that splicing of introns depends on this binding motif (Fig. 4I). Further, they provide evidence that the splicing defect leads to the pleiotropic phenotype of the Δ FgRbp1 mutant, by showing splicing defects in transcripts for ribosomal proteins, sexual developmental genes, and virulence factors. Overall, this manuscript covers a highly interesting, highly conserved topic and provides evidence for distinct functions of a conserved protein of the eukaryotic splicing machinery. Although the conclusions drawn from the results seem justified, I have major concerns about the presentation and clarity of the data. I would therefore suggest careful rewriting of the manuscript.

Comment 1: The manuscript contains many data that have been organized into 6 figures and 9 supplementary figures as well as supplementary tables and datasets. Several supplemental figures describing results are only mentioned in the discussion, although they represent results that have not been covered in the results section. For instance, the part on the SAGA complex represents results; however, to me these results on *gen5* seem much more preliminary than the results on FgRbp1 and the conclusions therefore speculative. This part should be either clarified and put in the results section or it should be omitted from the manuscript. Due to the SAGA part in the discussion, the discussion in total is rather short.

Re: Thanks for your good suggestions. We deleted the part on the SAGA complex and substantially revised our discussion in this revision.

Comment 2: The authors estimate the intron retention rate by comparing reads mapping to intronic versus exonic regions. However, this approach does not take into account sequencing bias. I would suggest using a more substantial approach for estimating splicing efficiency, e.g. calculating PIR (PIR was calculated as the percentage of the average number of reads mapping to the 5' and 3' exon–intron junctions, over the sum of the average of the exon–intron junction reads plus the exon–exon junction reads; from Braunschweig et al. 2014 Genome Res 24:1774–1786)

Re: We carefully read the paper [Braunschweig et al. 2014 Genome Res 24:1774–1786] that you recommended. PIR (percent intron retention) is an unbiased and elaborate method for calculating intron retention, one of the forms of alternative splicing. Although the fungus *F. graminearum* contains 77% intron-bearing genes, 98%

of them are constitutively spliced⁶ (only 231 genes in *F. graminearum* are subject to alternative splicing events), which is quite different from human genome in which extensive alternative splicing events are observed (>95% genes were alternative spliced). Therefore, the intron retention in *F. graminearum* is different from the intron retention in alternative splicing.

For comparison and visualization of splicing efficiency, we used the method that determine intron retention rate (or intron retention level) by calculating the ratio of reads aligned to intron to the reads aligned to overall exons for each tested intron, mainly referring to several articles previously reported in fungi^{7,8,9}.

Comment 3: It is often unclear to me, especially in first reading, how an analysis was performed and how the controls were done, how many introns a transcript contains and how many were tested (this information is also often lacking in the schematics provided in the figures).

Re: We added additional information to address these concerns in the revised main text as well as figure legends. For details, see below:

Comment 3-1: Strain designation Δ FgRbp1-C: Maybe I missed it, but what strain is this?

Re: Δ FgRbp1-C represents the complemented strain Δ FgRbp1::FgRbp1-GFP generated by transformation of Δ FgRbp1 with the full-length FgRbp1 protein fused with GFP. To be consistent with the text in Fig 1 and other contents in the manuscript, we have corrected “ Δ FgRbp1-C” to “ Δ FgRbp1::FgRbp1-GFP” in this revision.

Comment 3-2: Does the Rbp1-FLAG construct used for affinity capture complement the mutant phenotype?

Re: The strain Δ FgRbp1::FgRbp1-FLAG used for RIP or affinity capture completely restores the growth defect in Δ FgRbp1 (as shown in the following image).

Comment 3-3: Fig. 2: What is positive and negative control in Fig. 2C? Where is the control for the pGADT7-FgU2AF23 construct? What dilutions were plated? Similar

for further yeast two hybrid analysis in this manuscript

Re: A pair of plasmids, pGBKT7-53 and pGADT7-T, was used as the positive control, while pGBKT7-Lam and pGADT7-T was used as the negative control (see the figure legends in Fig 2B). The pGADT7-FgU2AF23 construct was used a control in the Fig 2B. The additional information for the supplemental figures (Fig S6 and S9) were also provided as suggested.

Comment 3-4: Fig. 2E: Please explain how you can distinguish YFP and RFP with an epifluorescence microscope?

Re: The laser excitation wavelength is different for RFP (red fluorescence) and YFP (yellow fluorescence). The laser excitation wavelengths for RFP and YFP were 561 nm and 514 nm, respectively.

Comment 3-5: Please describe in more detail how distinct transcripts were tested for splicing efficiency. Line 577, “spliced and unspliced primers”, is misleading.

Re: We have revised “measurement of splicing ratio” in Methods to provide clarification (lines 715-728). Also, schematic representation of primers used for amplifying spliced and unspliced mRNAs for a specific intron is shown in Fig S12.

Comment 3-6: For RNA-seq and RIP-seq, were the same growth conditions used? Please define “fresh mycelia” in line 520.

Re: The same growth conditions were used for RNA-seq and RIP-seq. As suggested, we have defined “fresh mycelia” in the revised manuscript in lines 644-646.

Comment 3-7: Which genome version were the sequencing data mapped to? Which genome version do the Locus tag numbers refer to?

Re: The genome version used in this study is *Fusarium graminearum* PH-1 from the website: https://www.ncbi.nlm.nih.gov/genome/58_genome_assembly_id=284608. The Locus tag numbers refer to the genome were downloaded from the website https://www.ncbi.nlm.nih.gov/genome/58_genome_assembly_id=284608.

Comment 3-8: Information on how the phylogenetic analysis was performed is missing. Is the tree in Fig. S3 based on the alignment in Fig. S4? If so, which part?

Re: The phylogenetic tree in Fig S3 was constructed based on the amino acid sequences of FgRbp1 and its orthologs in other organisms with Mega 5.0 using the neighbor-joining method. We have added the information in the figure legend for Fig

S3. The tree in Fig. S3 was based on the alignment in Fig. S4. The tree was constructed based on the alignment of the RRM domain indicated in Fig S4.

Comment 3-9: Data availability statement is missing

Re: We have added the Data availability statement in lines 759-763.

Comment 3-10: Line 569, define “fresh mycelia”

Re: We have defined “fresh mycelia” as suggested.

Comment 3-11: Line 233, “induced by these RPGs”, should this read “induced IN these RPGs”?

Re: As suggested, the sentence “induced by these RPGs” was edited to “induced in these RPGs”.

Comment 3-12: Line 319, “both the first and second intron of FgSTE12, reads like there are more than two introns, but legend to Fig. 6 implies there are only two.

Re: The gene *FgSTE12* contains two introns (as shown in Fig S7F).

Comment 3-13: I would suggest putting Fig. S7A together with Fig. 4D, since this is one central result of the manuscript

Re: As suggested, we have added the panel in Fig S7A to the Fig 5A in the revised manuscript.

Comment 3-14: The legends need to be revised to be more clear about what is shown and what has been done. For example, legend to Fig. S7A is much clearer than legend to Fig. 4D, which I think is wrong (this is a qRT-PCR analysis of the splicing, not the transcript level)

Re: We have substantially revised the figure legends in the revised manuscript.

Comment 3-15: When reading the manuscript, I am confused reading about conidiation assays in the Methods section (line 424), but not finding conidiation assays in the results. There are several more examples where methods and results description or legends do not match. For example, the legend to Fig. 2 mentions epifluorescence microscopy, while the methods section describes confocal microscopy (which would also make more sense)

Re: We have substantially revised the text in Results and Methods sections. We have also corrected the epifluorescence microscopy to confocal microscopy in Fig 2F.

Comment 3-16: Fig. S2, labeling in C not consistent with legend. Figure needs more explanation / schematic representation needs more information.

Re: We have revised the figure legends for Fig S2C.

Comment 3-17: Error bars are not always defined in the legends.

Re: We have defined the error bars in the figure legends.

Comment 3-18: Line 835-836, “inoculation... at 25 °C”, this does not make sense

Re: We have edited the sentence. Infected corn silks were examined 5 d after inoculation with a 5 mm mycelial plug of each strain. The information was added in the figure legends for Fig 7A.

Comment 4: The manuscript should be proofread by a native speaker, since there are a number of small mistakes

Re: A native speaker has proofread and thoroughly edited the manuscript.

References cited in this cover letter

1. Fu XD, Ares M. Context-dependent control of alternative splicing by RNA-binding proteins. *Nat Rev Genet* **15**, 689-701 (2014).
2. Wei W-J, *et al.* YB-1 binds to CAUC motifs and stimulates exon inclusion by enhancing the recruitment of U2AF to weak polypyrimidine tracts. *Nucleic Acids Res* **40**, 8622-8636 (2012).
3. Subramania S, *et al.* SAM68 interaction with U1A modulates U1 snRNP recruitment and regulates mTor pre-mRNA splicing. *Nucleic Acids Res* **47**, 4181-4197 (2019).
4. Wu SP, Romfo CM, Nilsen TW, Green MR. Functional recognition of the 3' splice site AG by the splicing factor U2AF(35). *Nature* **402**, 832-835 (1999).
5. Yoshida H, *et al.* A novel 3' splice site recognition by the two zinc fingers in the U2AF small subunit. *Genes Dev* **29**, 1649-1660 (2015).
6. Zhao CZ, Waalwijk C, de Wit P, Tang DZ, van der Lee T. RNA-Seq analysis reveals new gene models and alternative splicing in the fungal pathogen *Fusarium graminearum*. *BMC Genomics* **14**, 16 (2013).
7. Gao XL, *et al.* FgPrp4 Kinase Is Important for Spliceosome B-Complex Activation and Splicing Efficiency in *Fusarium graminearum*. *PLoS Genet* **12**, 23 (2016).
8. Sun M, Zhang Y, Wang Q, Wu C, Jiang C, Xu J-R. The tri-snRNP specific protein FgSnu66 is functionally related to FgPrp4 kinase in *Fusarium graminearum*. *Mol Microbiol* **109**, 494-508 (2018).
9. Kellner N, Heimel K, Obhof T, Finkernagel F, Kamper J. The SPF27 Homologue Num1 Connects Splicing and Kinesin 1-Dependent Cytoplasmic Trafficking in *Ustilago maydis*. *PLoS Genet* **10**, 20 (2014).

REVIEWER COMMENTS

Reviewer #1 (Remarks to the Author):

This manuscript by Wang and colleagues consist in a resubmission of a previously rejected study. The author have considerably amended their text and added new data to the precedent set of experiments. In this new version, the authors have addressed most points raised by the referee during the first evaluation round.

As it is, this version of the manuscript has been drastically improved with respect to its precedent version.

Results are sound and conclusions well supported by the experimental evidences presented. This manuscript appears suitable for publication in Nature Communications.

Remark

Fig.2E, is there any control in the author's hands showing that the RNase A treatment was good enough to abolish RNA mediated protein-protein interaction? It might be nice to include one.

<Reviewer #1's comment on the Reviewer #2's comment 2>

When it comes to IR calculation, the authors are citing in the Material and method section this manuscript:

Kellner N, Heimel K, Obhof T, Finkernagel F, Kamper J. The SPF27 homologue Num1 connects 805 splicing and kinesin 1-dependent cytoplasmic trafficking in *Ustilago maydis*. *PLoS Genet* 10, 806 20 (2014).

Strangely enough, while citing this manuscript they do not follow the IR calculation method described.

In this manuscript, Kellner and colleagues are calculating the IR as follow:

"Intron retention was determined by calculating the ratio of introns FPKM to the gene's overall exon FPKM. Exon FPKM included all exons with more than 10 bp. Only genes with an FPKM above 10 were considered".

While the calculation per se is similar, they do not use reads, as Minhui Wang and colleagues do, but FPKM (Fragments Per Kilobase Million).

This makes a big difference. Indeed, FPKM correspond to the number of "reads", here fragments, normalized for sequencing depth and gene length. With paired-end RNA-seq, two reads can correspond to a single fragment, or, if one read in the pair did not map, one read can correspond to a single fragment. Similarly, they could use RPKM (Reads Per Kilobase Million).

Thus, by using FPKM instead of reads, the authors would normalize their data sets for biases like sequencing depth and gene lengths.

I believe that, by strictly following the method used in Kellner et al., --i.e. use FPKM or RPKM-- the authors would satisfactorily address reviewer's #2 comment 2.

Alternatively, more sophisticated packages are also available like IRFinder ; IRFinder: assessing the impact of intron retention on mammalian gene expression, *Genome Biology* volume 18, Article number: 51 (2017)

An excellent description is provided with the code. This may also be good possibility.

Point-by-point response to reviewers' comments

Reviewer #1 (Remarks to the Author):

This manuscript by Wang and colleagues consist in a resubmission of a previously rejected study. The author have considerably amended their text and added new data to the precedent set of experiments. In this new version, the authors have addressed most points raised by the referee during the first evaluation round. As it is, this version of the manuscript has been drastically improved with respect to its precedent version. Results are sound and conclusions well supported by the experimental evidences presented. This manuscript appears suitable for publication in Nature Communications.

Response: Thank a lot for the positive comment.

Remark

Fig.2E, is there any control in the author's hands showing that the RNase A treatment was good enough to abolish RNA mediated protein-protein interaction? It might be nice to include one.

Response: Thanks a lot for the suggestion. The efficiency of RNase A treatment was detected when we conducted the Co-IP assay in Fig. 2e. The sensitivity of the FgRbp1-FgU2AF23 interaction to RNase A was determined by treating the lysates with RNase A (500 µg/ml) at room temperature for 30 min prior to immunoprecipitation and during immunoprecipitation at 4°C for 6h. After immunoprecipitation at 4°C for 6h, 1/10 volume (100 µl) of the lysate of each strain was saved and mixed with 1mL Trizol reagent for RNA extraction, while the remaining lysate together with the GFP-Trap agarose was subjected to washing and eluted for western blotting detection. The results showed that the RNase A treatment was effective (as shown in the following image). We have added the results of RNase A treatment in Fig. 2e (lower panel).

Reviewer #1's comment on the Reviewer #2's comment 2

When it comes to IR calculation, the authors are citing in the Material and method section this manuscript: Kellner N, Heimel K, Obhof T, Finkernagel F, Kamper J. The SPF27 homologue Num1 connects 805 splicing and kinesin 1-dependent cytoplasmic trafficking in *Ustilago maydis*. *PLoS Genet* 10, 806 20 (2014).

Strangely enough, while citing this manuscript they do not follow the IR calculation method described. In this manuscript, Kellner and colleagues are calculating the IR as follow: "Intron retention was determined by calculating the ratio of introns FPKM to the gene's overall exon FPKM. Exon FPKM included all exons with more than 10 bp. Only genes with an FPKM above 10 were considered". While the calculation per se is similar, they do not use reads, as Minhui Wang and colleagues do, but FPKM (Fragments Per Kilobase Million). This makes a big difference. Indeed, FPKM correspond to the number of "reads", here fragments, normalized for sequencing depth and gene length. With paired-end RNA-seq, two reads can correspond to a single fragment, or, if one read in the pair did not map, one read can correspond to a single fragment. Similarly, they could use RPKM (Reads Per Kilobase Million). Thus, by using FPKM instead of reads, the authors would normalize their data sets for biases like sequencing depth and gene lengths. I believe that, by strictly following the method used in Kellner et al., --i.e. use FPKM or RPKM-- the authors would satisfactorily address reviewer's #2 comment 2.

Alternatively, more sophisticated packages are also available like IRFinder; IRFinder: assessing the impact of intron retention on mammalian gene expression, Genome Biology volume 18, Article number: 51 (2017). An excellent description is provided with the code. This may also be good possibility.

Response: Thank you very much for your insightful suggestion. We have normalized the read counts to FPKM (Fragments Per Kilobase Million) for the respective intron/exon detected. After normalization, the intron retention rate was recalculated by using the ratio of introns FPKM to the gene's overall exon FPKM (as shown in Supplementary Data 3). Based on the recalculated intron retention rate, relevant figures and dataset have been revised (including Fig. 3a, 3b, 3d, 3e, Supplementary Fig. 10 and Supplementary Data 3) and the corresponding text contents have also been revised in the lines 199-208 and line 226-227. Meanwhile, the analysis method of intron retention rate has been revised accordingly in the method part in lines 658 to 665.

In general, the recalculated retention rates of introns using FPKM are consistent to the previous retention rates using the read counts. In the revised manuscript, 8689 introns in Δ FgRbp showed the intron retention rate 2-fold higher than those in the wild type (corresponding to 4849 genes that are defined as intron retention genes). Similarly, 8780 introns (corresponding to 4933 genes) were retained in the previous manuscript. The retained introns and corresponding intron-retention genes were highly overlapped in the two calculation assays (as shown in the following figure). In addition, when we overlapped FgRbp1-bound mRNAs (219) with the intron-retention mRNAs recalculated (4849), 170 out of 219 FgRbp1-bound mRNAs (78%) have the intron splicing defect in their pre-mRNAs (revised Fig. 3d), which is also similar to the previous overlapped genes (165 out of 219, 75%). Together, these results suggest that the conclusions based on the FPKM assay are in agreement with the previous conclusions.

A

8780 retained introns
(calculation with read counts)

8689 retained introns
(calculation with FPKM)

B

4933 intron-retention genes
(calculation with read counts)

4849 intron-retention genes
(calculation with FPKM)

REVIEWERS' COMMENTS

Reviewer #1 (Remarks to the Author):

In its revised version, this manuscript is now suitable for publication in Nature Communications.